# TANGO6 regulates cell proliferation via COPI vesicle-mediated RPB2 nuclear entry

Zhi Feng[1,8], Shengnan Liu[2,8], Ming Su[1], Chunyu Song [2], Chenyu Lin[2], Fangying Zhao[2], Yang Li [1], Xianyan Zeng[3], Yong Zhu[3], Yu Hou[3], Chunguang Ren [3], Huan Zhang[3], Ping Yi[4], Yong Ji [5,6], Chao Wang [7], Hongtao Li[2], Ming Ma, Lingfei Luo[2]✉ & Li Li [1]✉

Coat protein complex I (COPI) vesicles mediate the retrograde transfer of cargo between Golgi cisternae and from the Golgi to the endoplasmic reticulum (ER). However, their roles in the cell cycle and proliferation are unclear. This study shows that TANGO6 associates with COPI vesicles via two transmembrane domains. The TANGO6 N- and C-terminal cytoplasmic fragments capture RNA polymerase II subunit B (RPB) 2 in the *cis*-Golgi during the G1 phase. COPI-docked TANGO6 carries RPB2 to the ER and then to the nucleus. Functional disruption of TANGO6 hinders the nuclear entry of RPB2, which accumulates in the cytoplasm, causing cell cycle arrest in the G1 phase. The conditional depletion or overexpression of TANGO6 in mouse hematopoietic stem cells results in compromised or expanded hematopoiesis. Our study results demonstrate that COPI vesicle-associated TANGO6 plays a role in the regulation of cell cycle progression by directing the nuclear transfer of RPB2, making it a potential target for promoting or arresting cell expansion.

The RNA polymerase II complex is a fundamental part of the transcription apparatus[1], and its dysregulation leads to various disorders, including developmental deficiencies and tumors[2–4]. The functional RNA polymerase II complex encompasses 12 subunits, the largest of which are RNA polymerase II subunits B1 and B2 (RPB1 and RPB2)[5]. These subunits are produced in the cytoplasm and then move to the nucleus to play vital roles in the formation of the complete RNA polymerase II complex[6]. GTPase members such as GPN loop GTPase (GPN) 1, GPN3, RNA Polymerase II Associated Protein 2 (RPAP2), and Interacting with RNA polymerase II protein 1 (IWR1) are involved in the nuclear import of RPB1[7–9]. However, information regarding the nuclear shuttling of other subunits, particularly RPB2, remains elusive.

Coat protein complex I (COPI) vesicles are components of the endomembrane system in eukaryotic cells that precisely sort and carry cargo between the Golgi cisternae and from the Golgi to the endoplasmic reticulum (ER) via retrograde transport[10]. COPI coat assembly is initiated in the Golgi by the membrane recruitment of a heptamer coatomer and dynamic activation of small GTPases by ADP-ribosylation factor (ARF)-guanine exchange factors[11]. Cargo proteins are concomitantly recruited and contained with the assistance of

[1]Research center of Stem cells and Ageing, Chongqing Institute of Green and Intelligent Technology, Chinese Academy of Sciences, Chongqing 400714, PR China. [2]Institute of Developmental Biology and Regenerative Medicine, Key Laboratory of Freshwater Fish Reproduction and Development, Ministry of Education, Southwest University, Chongqing 400715, PR China. [3]Institute of Life Sciences, Laboratory of Developmental Biology, Department of Cell Biology and Genetics, Chongqing Medical University, Chongqing 400016, PR China. [4]Department of Obstetrics and Gynecology, The Third Affiliated Hospital of Chongqing Medical University, Chongqing 401120, PR China. [5]Key Laboratory of Cardiovascular and Cerebrovascular Medicine; Key Laboratory of Targeted Intervention of Cardiovascular Disease; Collaborative Innovation Center for Cardiovascular Disease Translational Medicine, Nanjing Medical University, Nanjing 211166, PR China. [6]National Key Laboratory of Frigid Zone Cardiovascular Diseases (NKLFZCD), Harbin Medical University, Harbin 150076 Heilongjiang, PR China. [7]MOE Key Laboratory for Membraneless Organelles & Cellular Dynamics, Hefei National Laboratory for Physical Sciences at the Microscale, School of Life Sciences, University of Science and Technology of China, Hefei 230027, PR China. [8]These authors contributed equally: Zhi Feng, Shengnan Liu. ✉e-mail: lluo@swu.edu.cn; lili@cigit.ac.cn

multiple matched sites on individual subunits[12]. Upon configuration, COPI vesicles transport cargo to the ER via a soluble N-ethylmaleimide-sensitive factor attachment protein receptor (SNARE) complex[13]. Disruption of COPI vesicle function arrests the expansion of lung cancer cells[14], although the underlying mechanism has not been elucidated.

The transport and Golgi organization (TANGO) initially designates a phenotypic class of genes showing defects in secretion upon knockdown in Drosophila S2 cells[15]. Although the roles of TANGO1/2 have been investigated in some detail[16–18], knowledge on the functions of TANGO6 is limited. In the present study, we found that TANGO6 was dynamically localized to the cytoplasm and nucleus during cell cycle progression. Cytoplasmic TANGO6 associates with COPI vesicles and recruits RPB2 via its N- and C-fragments at the G1 phase. Additionally, it carries RPB2 to the ER membrane via retrograde transport and delivers it to the nucleus with the assistance of leucine-rich repeat-containing 59 (LRRC59) and importins. Manipulation of the TANGO6-RPB2 axis arrests the proliferation of multiple cell types, including tumor cells in vitro and hematopoietic stem cells (HSCs) in vivo.

## Results

### TANGO6 acts as a specific COPI cargo

The mechanism underlying COPI and COPII vesicles regulation throughout the cell cycle remains unclear. Therefore, we characterized the patterns of COPI and COPII trafficking in sequential HeLa cell phases, an established system for cell cycle synchronization. The cells were collected at the G0, G1/S, S, and G2 phases via serum starvation[19] and double thymidine block assays[20]. These assays were verified using several cell cycle markers, including TK1 (S and G2 phase marker[21]) and cyclin A (G2 phase marker[22]) (Supplementary Fig. 1a). Immunofluorescence staining images indicated that GM130+ and GS28+ (cis-Golgi marker[23,24]), Giantin+ (mid-Golgi marker[25]), COPA+ (COPI vesicle marker[26]), and ERGIC53+ (ER-Golgi intermediate compartment marker[27]) signals were aggregated at the G0 and G1/S phases. The signals then diffused into the cytoplasm at the S phase and finally condensed again at the G2 phase (Fig. 1a and Supplementary Fig. 1b–e). However, the patterns of other organelles of the COP system, including the Sec31A-labeled COPII vesicle[16], TGN46, Golgin97-labeled trans-Golgi network[23,28], and Calnexin-marked ER[29] did not display cell-cycle-dependent localization alterations (Supplementary Fig. 1f–i). We carefully characterized the dynamics of GM130+ signals during S phase of the first seven hours[30]. The GM130+ signals condensed initially, which expanded one hour later. Subsequently, the GM130+ signals appeared to be isolated stacks and then disassembled into the Golgi hazes like structures, showing the highest total signal areas among the image periods. Eventually, the GM130+ signals re-converged again (Supplementary Fig. 1j–l). These results indicated the dynamic changes in the localization of cis-Golgi, mid-Golgi, and COPI vesicles throughout the cell cycle, which imply their involvement in carrying factors in cell proliferation. To investigate the roles of membrane trafficking factors in this process, 338 genes were initially examined via knockdown by delivering individual siRNAs (a mixture of three targets per gene). Three genes resulted in slightly expanded cell clones in U251 cells after knockdown; however, 223 genes resulted in reduced cell colonies compared to the controls (Supplementary Fig. 2a, b). The TANGO factors, including TANGO1, 2, 4, and 6, resulted in reduced cell numbers in this assay (Supplementary Fig. 2c, d). Compared to TANGO1, 2, and 4, information on TANGO6 is particularly limited.

Immunofluorescence staining data indicated that TANGO6, an approximately 130 KDa protein (Supplementary Fig. 2e), displayed a cytoplasmic dynamic localization similar to that of GM130+, Giantin+, GS28+ and COPA across the cell cycle (Fig. 1a and Supplementary Fig. 1b–d). In the S phase, the patterns of TANGO6+ and COPA+ signals were identical with those of GM130 (Supplementary Fig. 1m, n). TANGO6+ signals were predominantly detected in regions adjoining

the Golgi apparatus of GM130+ clusters (Fig. 1b), where TANGO6+ foci were aggregated and completely co-localized with COPA (Fig. 1c). Interestingly, certain sparsely dispersed puncta of TANGO6 were concomitantly observed in regions adjacent to the GM130+ cis-Golgi network (Fig. 1b). Approximately 23% of the scattered TANGO6 puncta co-localized with COPA+ signals (Fig. 1c, d). Super-resolution microscope-stimulated emission depletion imaging indicated that TANGO6+ puncta were located in the middle concave regions of COPA+ puncta (Fig. 1e). Co-immunoprecipitation (Co-IP) experiments indicated an interaction between TANGO6 and COPA/B (Supplementary Fig. 2f). After the knockdown of Arf1 and COPA (Supplementary Fig. 2g, h), aggregation intensity of TANGO6+ signals was reduced and discrete TANGO6+ signals were observed across the whole-cell cytoplasm (Supplementary Fig. 2i, j), but Golgi integrity was limitedly changed (Supplementary Fig. 2k). However, very little co-localization of TANGO6 with Sec31A was observed (Fig. 1c, d). These data indicate that TANGO6 largely resides on COPI but not COPII vesicles and that the interaction of TANGO6 with COPI vesicles is critical for its distribution. In contrast, immunofluorescence staining and western blot analysis results indicated that TANGO6 plays a limited role in the configuration of COPI vesicles and integrity of the Golgi apparatus (Fig. 1f, g and Supplementary Fig. 2l). Collectively, these results indicate that Golgi-derived COPI vesicles serve as crucial carriers of TANGO6 in the cytoplasm and are dynamically distributed during cell cycle progression (Fig. 1h).

### Cytoplasmic TANGO6 tethers to COPI vesicle via a signal anchor and two transmembrane domains

Subsequently, we investigated the mechanism underlying TANGO6 transport by COPI vesicles. We used several websites, including Phobius, Uniprot, Phyre2 and PSIPRED, to analyze the amino acid sequence of TANGO6. Six putative transmembrane (TM) domains and one signal anchor (SA) were predicted in the TANGO6 structure (Fig. 2a), based on a combination of the suggestions provided by all these websites. We recreated these elements by designing seven fused proteins linked by DsRed fragments. Immunofluorescence staining images indicated that the fused proteins containing the signal anchor (1–39 aa), TM2 (146–161 aa), and TM5 (472–487 aa), but not the other 4 elements, displayed localization patterns identical to those of TANGO6+ and COPA+ (Fig. 2b, c). However, omitting the signal anchor or the TM2 and/or 5 domains led to the diffusion of the TANGO6$^{\Delta SA/TMs}$-DsRed+ signals to the whole cytoplasm (Fig. 2d, e), thus validating the localization functions of these regions. We explored the cleavage signatures of the signal anchor in TANGO6 considering that signal anchors are largely cleaved for secretion. We constructed SP-HA-TANGO6(ΔSA) and HA-SP-TANGO6(ΔSA) plasmids containing the HA tag sequence after or before that of signal anchor (SA), followed by the TANGO6 fragments lacking SA (ΔSA) (Supplementary Fig. 3a). Western blot analysis indicated that HA activity was detected in SP-HA-TANGO6(ΔSA) or HA-SP-TANGO6(ΔSA) group. However, no similar appearance of HA was observed in the positive controls of HA-SP-IGF1(ΔSP) (Insulin-like growth factor 1, IGF1), whose signal peptide (SP) was cleaved[31] (Supplementary Fig. 3b). To further validate this result, we designed SA-HA-TANGO6(ΔSA)-FLAG construct to detect the patterns of HA and FLAG (Supplementary Fig. 3c). To this end, the cells were treated with digitonin and Triton X-100 to permeate the membranes of the cytoplasm and various organelles[16] (Supplementary Fig. 3d). We verified our assay using ERGIC53 as a positive control[32], which revealed that ERGIC53-N+ signals were exclusively detected in cells treated with Triton X-100, whereas ERGIC53-C+ signals were observed in both digitonin- and Triton X-100-treated cells (Supplementary Fig. 3e, f). The immunofluorescence images manifested a pattern of HA or FLAG that was compatible with that of endogenous TANGO6 after treatment by digitonin- and Triton X-100 (Supplementary Fig. 3g).

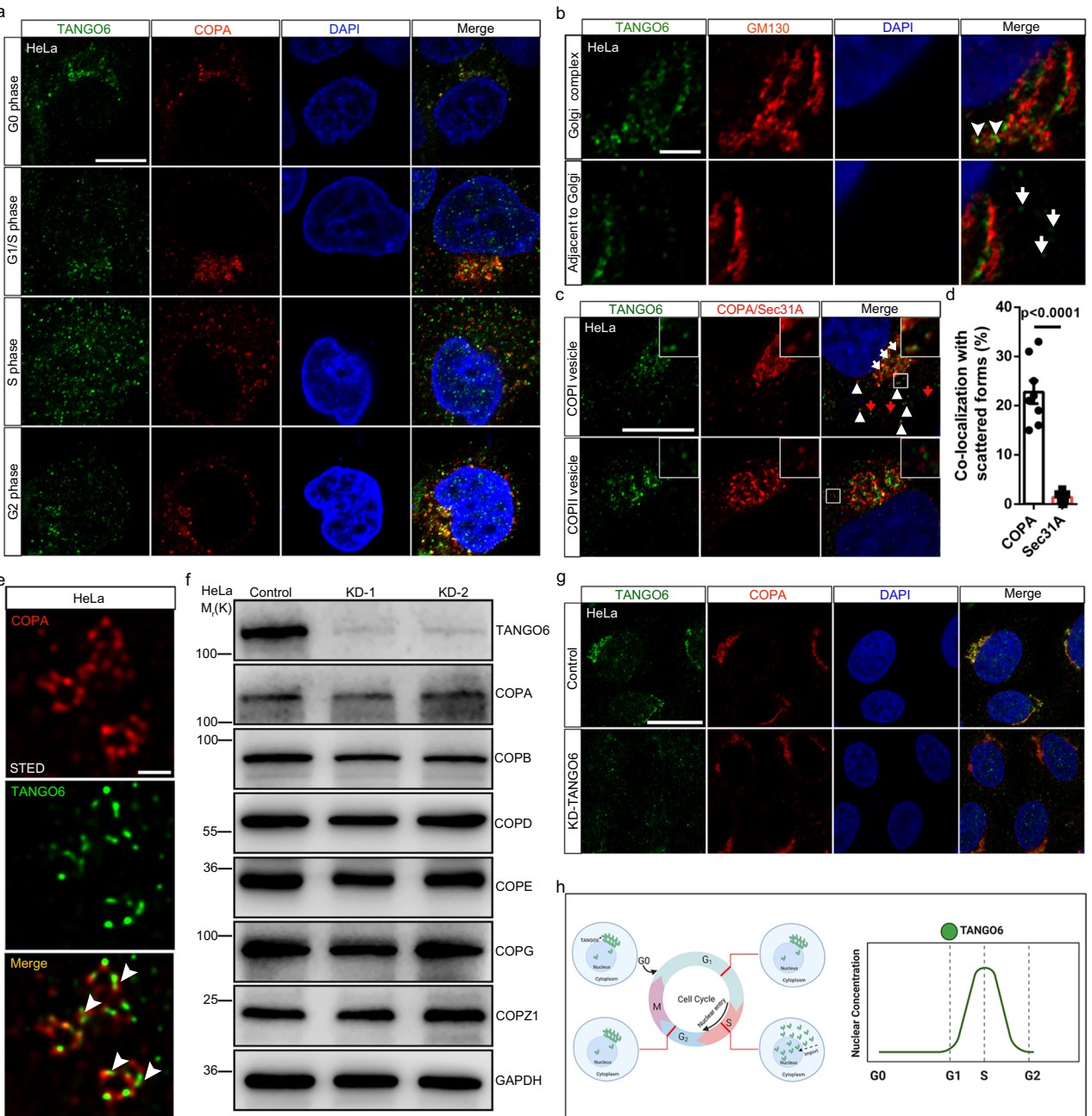

**Fig. 1 | The dynamic distributions of COPI-docked TANGO6 in cell cycle. a** The immunofluorescent staining images of TANGO6 and COPA in G0, G1/S, S and G2 phases of HeLa cells. Scale bar, 10 μm. **b** The immunofluorescent staining images of TANGO6 and GM130. The white arrowheads and arrows indicate the TANGO6⁺ dots adjoining (arrowheads) and adjacent (arrows) to the GM130⁺ clusters. Scale bar, 10 μm. The immunofluorescent staining images (**c**) and corresponding statistical results (**d**) of co-localization of TANGO6 with COPA and Sec31A. The up-right corners of each panel represent the enlarged images of the boxed regions. The white arrows and triangles indicate the co-localization of condensed and scattered forms of TANGO6 with COPA. The red arrows present the TANGO6 solely signals. (COPA, 22.75% ± 2.32%; Sec31A, 1.38% ± 0.38%. *n* = 8 cells in each group). Scale bar,

10 μm. **e** Super-resolution images of TANGO6 and COPA by STED. The white arrowheads indicate the merged signals. Scale bar, 200 nm. **f** Western blot analysis of COPI subunits (COPA/B/D/E/G/Z1) after knocking down TANGO6 by siRNA. GAPDH is the internal standard. KD, knock down. **g** The immunofluorescent staining images of COPA when transiently knocking down TANGO6 by siRNA. Scale bar, 20 μm. **h** The diagram of TANGO6 distribution (left) and nuclear concentration (right) during interphase of cell cycle (G0, G1, S, G2) progression. Statistical significance for (**d**) was assessed using unpaired one-tailed Student's *t* test. Mean ± s.e.m. The models were created with BioRender.com. Source data are provided as a Source Data file.

We then explored the topological localization of TANGO6 on the COPI vesicle by TM2 and/or 5 domains by referring to a study on TANGO1, which similarly harbors two TMs. We stained TANGO6 with an antibody against the C-terminus (599–694 aa). The TANGO6⁺ signals were co-stained with COPA in both the PFA- and PFA plus Triton X-100-treated groups (Supplementary Fig. 3h), implying that the

C-terminal fragment of TANGO6 was localized in the cytoplasm but not in the vesicle lumen. We designed an HA-TANGO6-FLAG plasmid for the detection of the N- and C- termini of TANGO6 using HA and FLAG antibodies, respectively (Supplementary Fig. 3i). Immuno-fluorescence staining images revealed that both HA⁺ and FLAG⁺ signals that merged with TANGO6 were observed in transfected cells treated

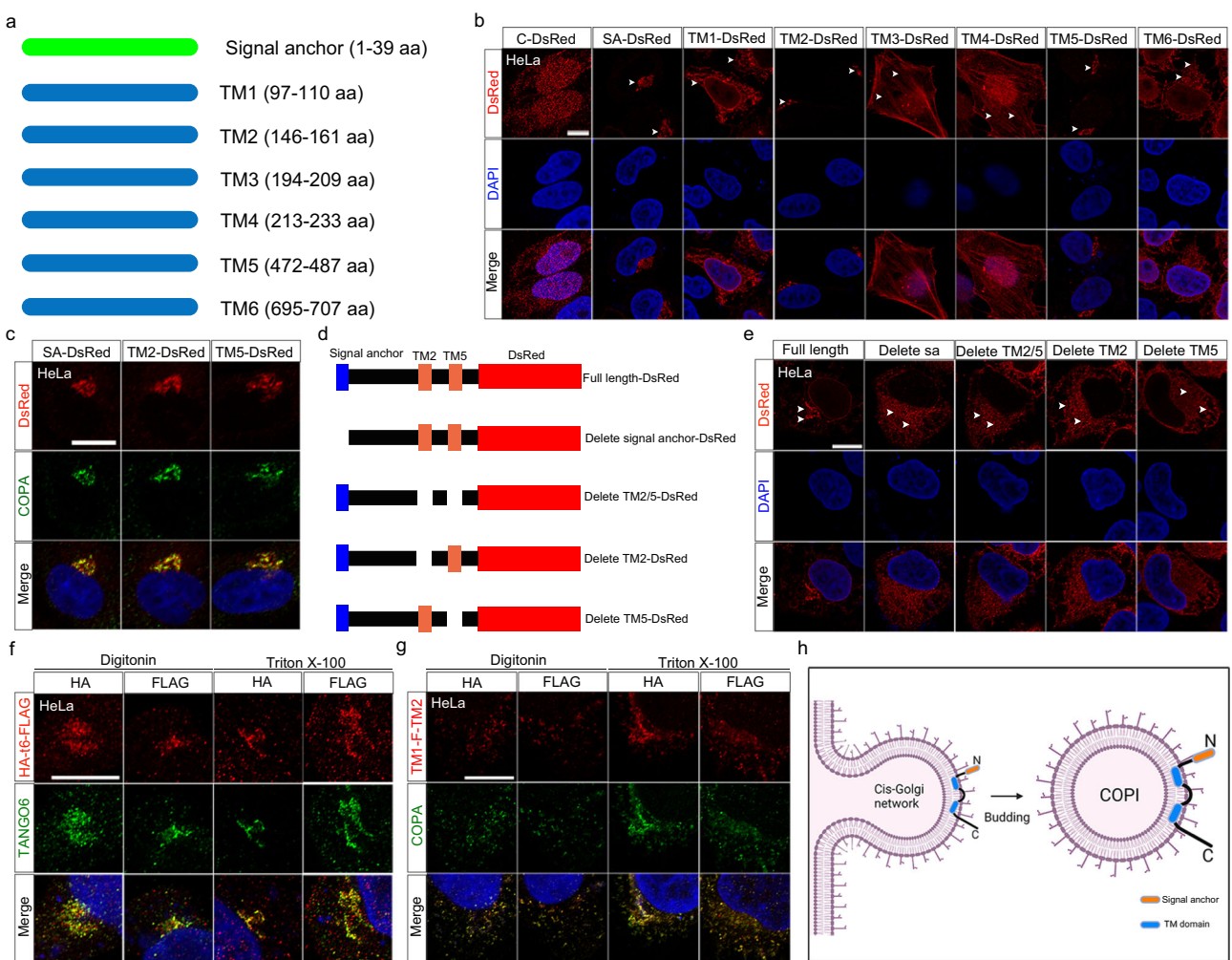

**Fig. 2 | The topology of TANGO6 in COPI vesicles. a** The diagram of signal anchor and transmembrane domains (TM). **b** The immunofluorescent staining images of DsRed fused signal anchor (SA) and TM elements. The white arrowheads indicate DsRed⁺ signals. C, Control. Scale bar, 10 μm. **c** The immunofluorescent staining images of co-localization of DsRed fused different elements (signal anchor, TM2 and TM5) with COPA. Scale bar, 10 μm. The diagram of omitting signal anchor, TM2 and TM5 elements (**d**) and confocal images of DsRed⁺ signals distribution after deleting corresponding fragment (**e**). The white arrowheads indicate DsRed⁺ signals. Scale bar, 10 μm. The immunofluorescent staining images of HA-TANGO6-FLAG and TANGO6 (**f**) or TM1-HA-F (fragments 161–472 aa)-FLAG-TM2 and COPA (**g**) treated by digitonin or Triton X-100. Scale bar, 10 μm. **h**, The diagram of TANGO6 topology on COPI vesicle after budding from *cis*-Golgi network. TANGO6 locates on the COPI vesicle via its signal anchor (orange) and two transmembrane domains (blue). The models were created with BioRender.com.

with either digitonin or Triton X-100 (Fig. 2f). These data indicated that the signal anchor was maintained and both the N- and C- termini of TANGO6 were localized at cytoplasm. Based on this, we proposed two feasible topologies with the segment 161–472 aa located either in the lumen or in the cytoplasm (Supplementary Fig. 3j). To weigh these models, we constructed a TM1-HA-F (fragment 161–472 aa)-FLAG-TM2 plasmid by fusing the HA tag with TM1 motif and FLAG tag with the TM2 motif. The immunofluorescence staining images revealed both HA⁺ and FLAG⁺ signals in either the digitonin or Triton X-100 treated groups (Fig. 2g), indicating that the fragment 161–472 aa was located in the cell cytoplasm. These findings supported that TANGO6 was tethered to COPI vesicles via the two transmembrane domains, and the left fragments were located in cytoplasm (Fig. 2h).

Previous research reported that TANGO1 regulated Collagen VII secretory[16], implying the involvement of TANGO6 in protein secretion. To this end, we collected the proteins from cell lysate and medium supernatants after knocking down TANGO6. EGF and Collagen I (the two typical secretory cargoes[33,34]) were examined. However, the results indicated that the protein levels of EGF and Collagen I were comparable between the cell lysate and supernatant upon TANGO6 knockdown in the HeLa cells (Supplementary Fig. 3k). To exclude the

possible effects of knockdown efficiency on this result, we completely disrupted the TANGO6 function by generating mutant cells using CRISPR/Cas9. Although we failed to obtain *TANGO6* homozygous mutants in HeLa cells, we fortunately obtained the *TANGO6* mutant U251 cell line (*TANGO6* ᴷᴼ). *TANGO6* ᴷᴼ is a heterozygous cell line with simultaneous deletions of 41 and 86 bp in each strand of the gene, which produces a protein with diminished activity (Supplementary Fig. 3l, m). We performed the pulse chase assay on these cells[35]. The immunofluorescent staining images indicated that both *TANGO6* ᵂᵀ and *TANGO6* ᴷᴼ cells contained alanine (Supplementary Fig. 3n). The relative abundance of [2,3-¹³C₂] alanine gradually reduced in the cell pellet and increased in the medium supernatant in the WT cell samples. However, no compromised secretion was detected in *TANGO6* ᴷᴼ samples. Interestingly, *TANGO6*ᴷᴼ cells presented a much faster decline of [2,3-¹³C₂]alanine in the cell pellet but quicker ascend in the medium supernatant during the first 2 h of detection than their WT counterparts (Supplementary Fig. 3o, p).

**TANGO6 enters the nucleus via the COPI- SNARE-ER pathway**
In addition to the dynamic distribution of TANGO6 in the cytoplasm, TANGO6⁺ signals were also observed in the nucleus, particularly during

the G1/S and S phases (Fig. 1a). We separated the cytoplasm from the nucleus during different cell phases. TANGO6 protein was easily detected in both the cytoplasm and nucleus by western blot analysis, although it was more enriched in the cytoplasm (Fig. 3a). Although the overall protein levels of TANGO6 were fairly consistent throughout the

cell cycle (Supplementary Fig. 4a), the intensity of nuclear TANGO6 was notably reduced at the G2 phase compared to other phases (Fig. 3a). Conversely, cytoplasmic TANGO6 displayed the opposite trend, with a notable increase during the G2 phase (Fig. 3a). This result implies the nuclear import of TANGO6. We tagged TANGO6 with

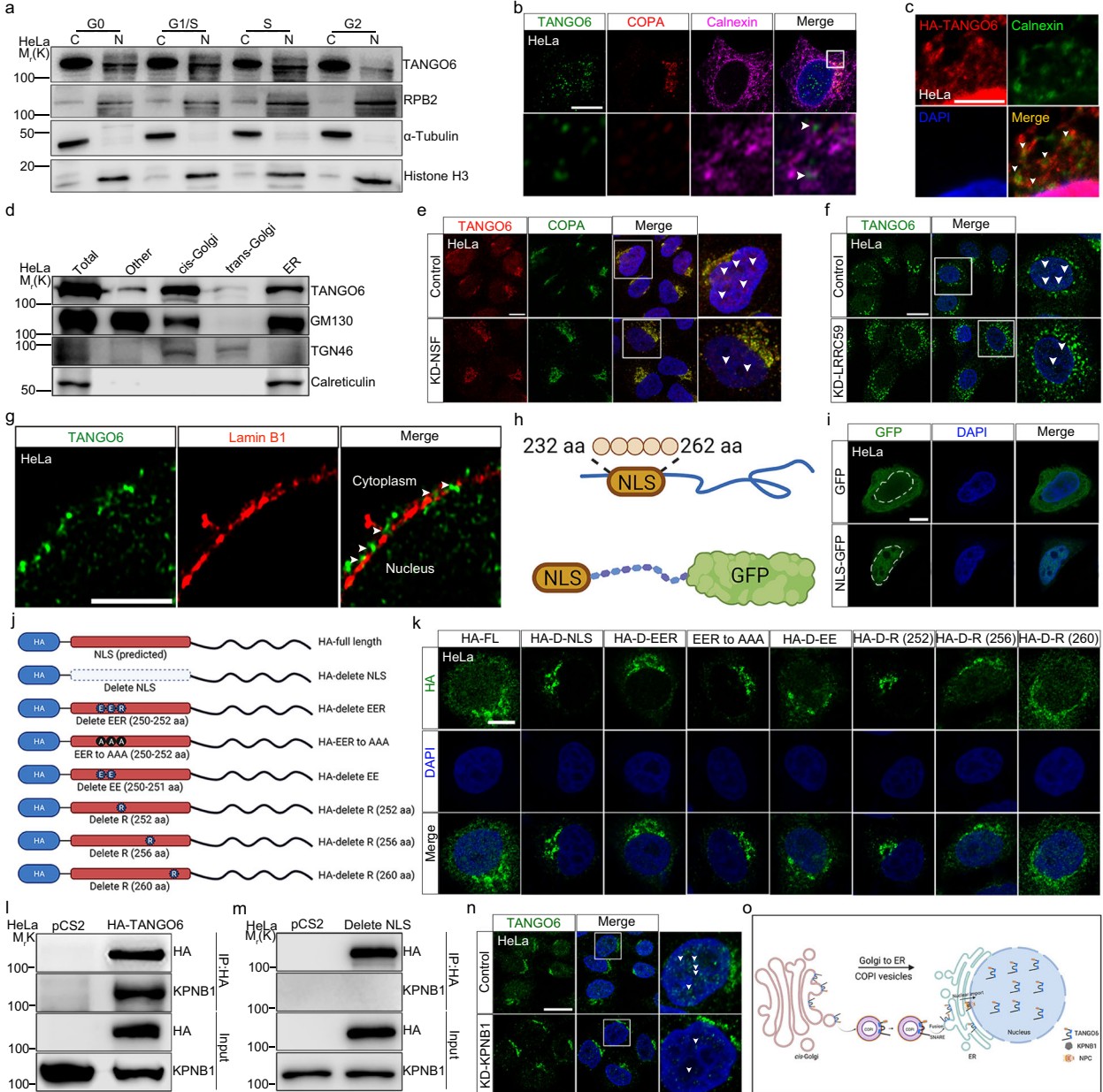

**Fig. 3 | The nuclear import of TANGO6 is assisted by SNARE complex, ER and KPNB1. a** Western blot analysis of TANGO6 and RPB2 distribution in cytoplasm (C) and nucleus (N) during different interphases of cell cycle. **b** The immuno-fluorescent staining of TANGO6, COPA and Calnexin. The bottom panels are the magnified field of boxed regions. The white arrowheads indicate the TANGO6⁺COPA⁻ dots co-localized with Calnexin⁺ signals. Scale bar, 10 μm. **c** The immunofluorescent staining images of HA-TANGO6 and Calnexin. The white arrowheads indicate merged signals. Scale bar, 10 μm. **d** Western blot analysis of TANGO6 in the fractions of *cis*-Golg, *trans*-Golgi and ER. ER, endoplasmic reticulum. **e** The immunofluorescent staining images of TANGO6 and COPA when knocking down NSF. Scale bar, 20 μm. **f** The immunofluorescent staining images of TANGO6 after knocking down LRRC59. The white arrowheads indicate nuclear TANGO6⁺ signals. Scale bar, 20 μm. **g** The immunofluorescent staining images of TANGO6 and Lamin B1. The white arrowheads indicate the TANGO6⁺ signals inside the space

between Lamin B1⁺ dots. Scale bar, 10 μm. The diagram of NLS fused with GFP (**h**) and the fluorescent images of GFP fused with or without NLS element (**i**). The white dashed lines indicate the nucleus. NLS, nuclear localization signal. aa amino acid. Scale bar, 10 μm. **j** The diagram of constructs with different NLS mutations. **k** The immunofluorescent staining images of HA after transfecting various constructs in **j**. FL full length. D delete. Scale bar, 10 μm. Co-immunoprecipitation of KPNB1 with TANGO6 (**l**) or TANGO6 ᐩᴺᴸˢ (delete NLS) (**m**). IP immunoprecipitation. **n** The immunofluorescent staining images of TANGO6 when knocking down KPNB1. The right panels are amplified images of boxed region. The white arrowheads indicate the TANGO6⁺ signals in nucleus, Scale bar, 20 μm. **o** The schematic diagram of TANGO6 nuclear import process. COPI vesicle dwelled TANGO6 is derived from *cis*-Golgi and moves to ER via the retrograde transport that is mediated by SNARE. The subsequent nuclear entry of TANGO6 is assisted by its NLS and KPNB1. The models were created with BioRender.com. Source data are provided as a Source Data file.

mMaple3, a photo-convertible fluorescent protein[36]. After photo-conversion, TANGO6-mMaple3 protein was observed as red in the cytoplasm. After 3 h, the red TANGO6-mMaple3$^+$ signals increased in the nucleus (Supplementary Fig. 4b), indicating the nuclear entry of cytosolic TANGO6.

We explored the mechanisms underlying TANGO6 nuclear translocation. COPI vesicles transport cargo to the ER. TANGO6$^+$ puncta that did not merge with the scattered COPA$^+$ signals in the cytoplasm (~75%) predominantly co-localized with Calnexin$^+$ clusters (Fig. 3b). HA-TANGO6 was transfected into the cells, and HA$^+$ signals were detected in the Calnexin$^+$ labeled ER (Fig. 3c). The Golgi complex and ER were then isolated using an organelle extraction kit. TANGO6 activity was observed in the *cis*-Golgi and ER fractions but was barely detectable in the *trans*-Golgi fragments (Fig. 3d). SNARE complex-mediated membrane fusion plays a crucial role in the retro-transport of cargo from the Golgi apparatus to the ER via COPI vesicle[37]. Co-IP results indicated that TANGO6 interacted with N-ethylmaleimide-sensitive fusion protein (NSF) (Supplementary Fig. 4c), a key ATP enzyme involved in SNARE complex-mediated membrane fusion[38]. These data implied that the COPI-SNARE-ER pathway was involved in the nuclear entry of TANGO6. This was supported by the observation that disruption of the COPI vesicle through knockdown of Arf1 and COPA (key components of COPI vesicles) caused significant reduction in nuclear TANGO6$^+$ signals (Supplementary Fig. 2i, j). Transient knockdown of NSF or thapsigargin (TA) treatment (to disrupt ER integrity[39]) led to a reduction in the TANGO6$^+$ nuclear signals and a concomitant increase in TANGO6$^+$ aggregation in the cytoplasm (Fig. 3e and Supplementary Fig. 4d–f). To explore the mechanism underlying nuclear entry of TANGO6 from ER, LRRC59, an ER resident protein that mediated nuclear import of membrane proteins[40], was examined. Co-IP results indicated considerable interaction between TANGO6 and LRRC59 (Supplementary Fig. 4g). The immunofluorescent staining and western blot analysis data indicated significant accumulation of TANGO6$^+$ signals in the cell cytoplasm but reduced intensities in the nucleus after knockdown of LRRC59 (Fig. 3f and Supplementary Fig. 4h–j). However, TANGO6 protein levels and ER integrity (marker by Calnexin) were not notably affected under this condition (Supplementary Fig. 4h, i). This indicated the LRRC59-mediated nuclear entry of TANGO6 from ER.

## Nuclear entry of TANGO6 is assisted by KPNB1

Integrated transmembrane proteins in the ER expose a nuclear localization sequence (NLS) element to bind importins, which then travel to the nucleus via the nuclear pore complex[40,41]. Using Lamin B1 to label the nuclear membrane[42] revealed that TANGO6$^+$ puncta were localized very close to the Lamin B1$^+$ signals, with some residing in the spaces between the adjunct Lamin B1$^+$ puncta (Fig. 3g). Concordantly, a candidate NLS was predicted to be located at the N-terminus (232–262 aa) of TANGO6. Transfection of a fused NLS-GFP plasmid revealed distinct GFP$^+$ signals in the cell nucleus compared to the GFP control group (Fig. 3h, i). Subsequently, we designed a series of HA-TANGO6 fused proteins with various mutations targeting key sites in the NLS elements (Fig. 3j). Upon modification of the whole NLS sequence or three amino acids (Glu/Glu/Arg) from the 250–252 site through either ablation or mutation (to Ala), HA$^+$ signals in the nucleus disappeared. However, when only two Glu sites were modified, these effects were limited (Fig. 3k). In contrast, when Arg at site 252, rather than at sites 256 or 260, was removed, the HA$^+$ signals resembled that observed after the deletion of Glu/Glu/Arg (Fig. 3k). Karyopherin-β family members interact with the NLS sequence of most macromolecules to facilitate their nuclear transport[43]. TANGO6 interacted and partly co-localized with karyopherin subunit beta-1 (KPNB1, an importin β[44]) (Fig. 3l and Supplementary Fig. 4k), and ablation of the TANGO6 NLS disrupted this interaction (Fig. 3m). KPNB1 knockdown caused drastic reduction in TANGO6$^+$ signals in the nucleus and the concomitant accumulation

of TANGO6$^+$ in the cytoplasm (Fig. 3n and Supplementary Fig. 4l, m). In summary, these results indicate that TANGO6 is carried to the ER by COPI vesicles and enters the nucleus with the assistance of LRRC59 and KPNB1. The NLS element of TANGO6, particularly the Arg at position 252, plays a vital role in its nuclear translocation (Fig. 3o).

## TANGO6 carries RPB2 to the nucleus

Our study results indicate the nuclear entry of TANGO6. However, DNase I treatment of the nuclear preparation[36] did not increase the amount of TANGO6 in the nucleoplasm (Supplementary Fig. 5a), implying that TANGO6 is not bound to chromatin and may therefore be involved in the transport of various factors to the nucleus. Therefore, we conducted mass spectrum analysis to identify candidate molecules. Glioma cell line U251 was utilized because TANGO6 is highly enriched in glioma compared to cervical carcinoma (http://gepia2.cancer-pku.cn/#general). We confirmed the conserved co-localization of TANGO6 with COPA and the similar cytoplasmic and nuclear distributions of TANGO6 in U251 cells (Supplementary Fig. 5b, c). We used TANGO6 or FLAG antibodies to pull down endogenous or over-expressed FLAG-tagged TANGO6 proteins, respectively (Supplementary Fig. 5d). Mass spectrometry results from both assays revealed that RPB2, the second-largest subunit of the RNA polymerase II complex, displayed substantial interaction with TANGO6 (Fig. 4a). Subsequent Co-IP experiments confirmed the strong interaction between TANGO6 and RPB2 (or vice versa) in HeLa and U251 cells (Fig. 4b and Supplementary Fig. 5e, f). However, no interaction was observed between TANGO6 and RPB1, the largest subunit of RNA polymerase II complex[5] (Fig. 4b). These data suggest that TANGO6 is involved in the transport of RPB2 to the nucleus. Electron and confocal microscopy revealed integrity of the Golgi structure and COPI vesicles that were comparable between *TANGO6*$^{KO}$ and WT cells (Supplementary Fig. 5g, h). However, *TANGO6*$^{KO}$ U251 cells revealed drastic reduction of RPB2$^+$ in the nucleus, concomitant with a marked aggregation in the cytoplasm (Fig. 4c). Interestingly, the assembly of RPB2$^+$ foci in the cytoplasm of *TANGO6*$^{KO}$ cells displayed ball-like structures of various sizes (Fig. 4c); however, RPB2 activity, and not that of RPB1, decreased obviously (Supplementary Fig. 5i). A similar phenomenon was observed in TANGO6 knockdown (KD) in HeLa and U251 cells (Fig. 4d and Supplementary Fig. 5j, k). Moreover, when HA-tagged RPB2 was transfected into *TANGO6*$^{KO}$ cells, it accumulated in the cytoplasm but did not enter the nucleus (Supplementary Fig. 5l). We extensively examined the signatures of other 11 Pol II subunits in this scenario. The immunofluorescence staining results indicated that the suppression of TANGO6 had no discernable effect on the nuclear distributions of these 11 Pol II subunits (Supplementary Fig. 6b–d); however, the protein levels of RPB3/6/11 decreased simultaneously (Supplementary Fig. 6a).

## De novo TANGO6 captures RPB2 via N- and C-terminal fragments at the G1/S phase

Upon finding that TANGO6 regulated the nuclear transport of RPB2, we explored how RPB2 is captured and carried by TANGO6. Separation of the cytoplasm and nucleus followed by Co-IP assays indicated that the interaction between TANGO6 and RPB2 predominantly occurred in the cytoplasm (Fig. 4e and Supplementary Fig. 5m). Consistently, we observed dominant co-localization of cytoplasmic TANGO6 with RPB2 in immunofluorescence experiments; however, certain RPB2$^+$ puncta did not co-localize with TANGO6 (Fig. 4f). To elucidate the location of the TANGO6 -RPB2 interaction, sucrose density gradient centrifugation was performed to separate cells into distinct compartments according to their sedimentation coefficients[45]. TANGO6, RPB2, and COPA were enriched in the second fraction, which contained relatively few other proteins including RPB1, Sec31A, ERGIC53, GM130, and Calnexin (Fig. 4g). COPI vesicles were derived from the *cis*-Golgi network[46]. The *cis*-Golgi and *trans*-Golgi components were separated.

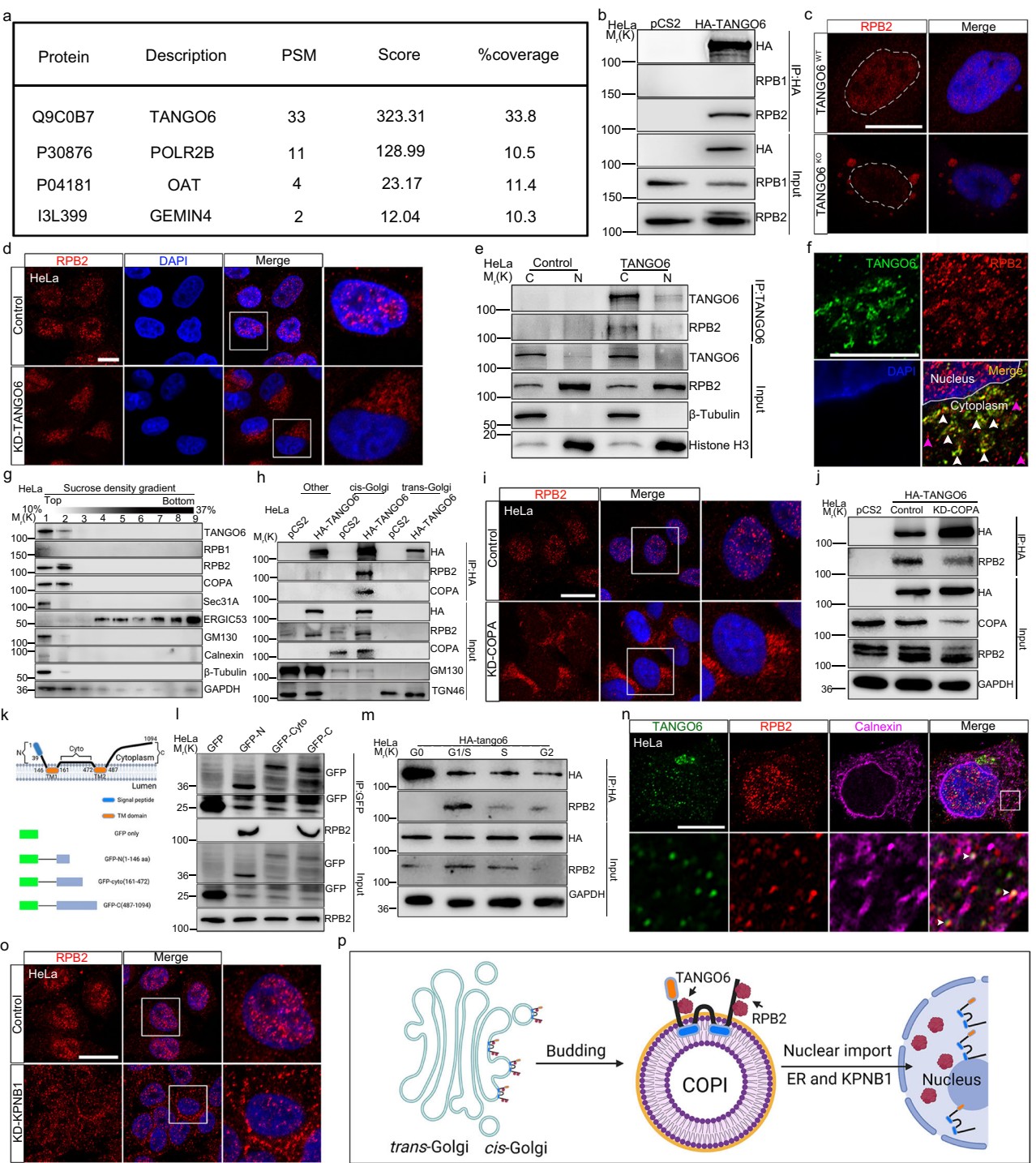

Co-IP results confirmed that TANGO6 predominantly interacts with RPB2 in the *cis*-Golgi network, where TANGO6 is enriched (Figs. 3d and 4h). No such interactions were observed in the *trans*-Golgi apparatus (Fig. 4h). We transiently knocked down COPA in HeLa and U251 cells using siRNA and observed that RPB2 was reduced in the nucleus but notably accumulated in the cell cytoplasm (Fig. 4i and Supplementary Fig. 5n), recapitulating the results observed in TANGO6-deficient cells. The interaction between TANGO6 and RPB2 was also significantly reduced (Fig. 4j).

We then divided TANGO6 into three parts based on the two TM domains and constructed GFP-tagged constructs to create the GFP-N-terminus (1–146 aa), GFP-cyto (161–472 aa), and GFP-C-terminus

(487–1094) (Fig. 4k). The Co-IP results indicated that the N- and C-termini of TANGO6 (GFP-N-/C- termini) interacted with RPB2, whereas cyto-TANGO6 failed to capture RPB2 (Fig. 4k, l). Considering the dynamic temporal distributions of COPI vesicles, TANGO6, and RPB2 proteins, we identified the cell cycle phases where TANGO6 interacts with RPB2. Therefore, we isolated proteins at different cell cycle phases of interphase to perform Co-IP experiments. The results indicated that the TANGO6-RPB2 interaction was the strongest at the G1/S phase, obviously reduced at the S and G2 phases, and barely measurable at the G0 phase (Fig. 4m). In addition, western blot analysis showed that the protein levels of nuclear RPB2 increased at the S and G2 phases compared to the G0 and G1/S phases (Fig. 3a), suggesting

**Fig. 4 | TANGO6 captures cytoplasmic RPB2 and carries it into nucleus. a** Mass spectrum results. Three candidate proteins, POLR2B (RPB2), OAT and GEMIN4, are identified from high to low scores. TANGO6 is used as a positive control. **b** Co-immunoprecipitation of TANGO6, RPB1 and RPB2 in HeLa cells. **c** The immuno-fluorescent staining images of RPB2 in *TANGO6*^WT and *TANGO6*^KO U251 cells. WT, wild type. KO, knock out. Scale bar, 10 µm. **d** The immunofluorescent staining images of RPB2 after knocking down TANGO6. Scale bar, 20 µm. **e** Co-immunoprecipitation of RPB2 with TANGO6 in cytoplasm (C) and nucleus (N). β-Tubulin and Histone H3 are used as internal standards of cytoplasmic and nuclear proteins respectively. **f,** The immunofluorescent staining images of TANGO6 and RPB2. The white and purple arrowheads indicate the co-staining and RPB2 solely signals respectively. Scale bar, 10 µm. **g** Western blot analysis of TANGO6, RPB1/2, COP vesicle (COPA, Sec31A and ERGIC53), *cis*-Golgi (GM130) and ER (Calnexin) in each sucrose gradient fraction (from 10% to 37%). **h** Co-immunoprecipitation of HA-TANGO6, RPB2 and COPA (COPI vesicle marker) after extraction of *cis*-Golgi and

*trans*-Golgi components. **i** The immunofluorescent staining images of RPB2 after knocking down COPA. Scale bar, 20 µm. **j** Co-immunoprecipitation of HA-TANGO6 and RPB2 when knocking down COPA. The diagram of GFP fused N- (1–146 aa) terminal (GFP-N), Cytoplasmic (161–472 aa, GFP-cyto) and C- (487–1094 aa) terminal (GFP-C) fragments (**k**) and Co-immunoprecipitation of TANGO6 different fragments with RPB2 (**l**). **m** Co-immunoprecipitation of HA-TANGO6 with RPB2 in different phases of cell cycle. **n** The immunofluorescent staining images of TANGO6, RPB2 and Calnexin. The down panels are the amplified images of boxed region. The white arrowheads indicate the merged signals of TANGO6, RPB2 with Calnexin. Scale bar, 10 µm. **o** The immunofluorescent staining images of RPB2 when knocking down KPNB1. Scale bar, 20 µm. **p** The diagram shows that COPI-docked TANGO6 captures RPB2 via its cytoplasmic N- and C- terminal fragments and directs it nuclear entry via the assistance of ER and KPNB1. The models were created with BioRender.com. Source data are provided as a Source Data file.

that TANGO6 captures RPB2 at the G1/S phase prior to nuclear entry. These data support the notion that COPI vesicle-associated TANGO6 captures RPB2 largely at the G1/S phase via its N- and C-terminal fragments.

As TANGO6 nuclear import is assisted by ER and KPNB1, we subsequently investigated the role of ER in RPB2 transport. We observed the co-localization of RPB2 and TANGO6 puncta in the Calnexin structure (Fig. 4n). The Co-IP assay demonstrated the interaction between TANGO6 and RPB2 in the ER fraction (Supplementary Fig. 5o). Knockdown of KPNB1 in HeLa cells led to a similar reduction in nuclear RPB2 and a concomitant increase in cytoplasmic RPB2 as observed with TANGO6 disruption (Fig. 4o). These results indicate that TANGO6 transports RPB2 to the nucleus via COPI vesicle-mediated trafficking, which is assisted by KPNB1 from the ER (Fig. 4p).

## TANGO6-RPB2 axis plays an indispensable role in cell proliferation

We investigated the significance of TANGO6-mediated RPB2 nuclear transport. We observed that *TANGO6*^KO cells grew considerably slower than WT cells (Fig. 5a). Examination of Ki-67, PCNA, and CDK1 by immunofluorescence staining or western blot analysis indicated that their levels were considerably lower in the *TANGO6*^KO nucleus than in the WT controls (Fig. 5b, c). However, the active Caspase-3+ signals did not show notable alterations (Supplementary Fig. 7a). Data from RNA sequencing (RNA-seq) analysis revealed that 491 and 918 genes were upregulated and downregulated, respectively, in *TANGO6*^KO versus WT cells (Supplementary Fig 7b). The transcription levels of RNA polymerase II subunits, including RPB2/3/4/10, were detectably reduced in *TANGO6*^KO cells (Supplementary Fig. 7c, d). Gene set variation analysis[47] revealed that the transcription-dependent tethering activity of RNA polymerase II was reduced (Supplementary Fig. 7e), whereas the biological activity of RNA polymerases I and III and secretory pathway remained unchanged in *TANGO6*^KO cells (Supplementary Fig. 7f–h). The processes of G1/S transition and DNA replication were mitigated in *TANGO6*^KO cells compared to controls (Fig. 5d, e), whereas no alterations were detected in the G2/M transition (Supplementary Fig. 7i). Consistently, the activity of Cyclin E (an indicator of the G1/S checkpoint[22]) was increased, whereas those of TK1 and Cyclin A decreased in *TANGO6*^KO cells compared to *TANGO6*^WT cells (Supplementary Fig. 7j). These data suggest that TANGO6 deficiency leads to cell cycle arrest at the G1/S phase, where the TANGO6-RPB2 interaction is strongest (Fig. 4m). Similarly, we performed rescue experiments by delivering an NLS-HA-PRB2 plasmid that contained an NLS[48] (Pro-Lys-Lys-Lys-Arg-Lys-Val) signal peptide at the N- terminus of HA-RPB2. Consequently, a clear nuclear appearance of NLS-HA-RPB2 was discerned (Supplementary Fig. 7k), thereby significantly recovering the Ki-67+ signals and cell colonies in the *TANGO6*^KO cells. However, the HA-RPB2 plasmid supplement presented limited effects (Fig. 5f). Consistently, knockdown of TANGO6 activity in HeLa

cells led to substantial reduction in Ki-67+ signals in the nucleus compared to the controls (Fig. 5g). Knockdown of Arf1 (a recruitment subunit of the COPI vesicle[49]), COPA, and COPB (the two key components of the COPI vesicle) by validated siRNA (Supplementary Fig. 2g, h, and 7l) caused cell growth arrest (Fig. 5h, i) and significant reduction in PCNA, Cyclin A, and Ki-67 levels in HeLa cells, according to both western blot analysis and immunofluorescence staining results (Fig. 5j, k). Moreover, application of TA to induce ER stress[39] or siRNA to transiently knockdown KPNB1 or RPB2 significantly suppressed cell proliferation (Supplementary Fig. 7m–s). Furthermore, we estimated the combinational roles in TANGO6, RPB2, and COPA. The protein levels of TK1 and Cyclin A and the intensities of Ki-67+ signals reduced significantly upon the simultaneous knockdown of any two factors (TANGO6/RPB2, TANGO6/COPA, and RPB2/COPA). However, simultaneous interfering RPB2 and COPA yielded the most conspicuous effects (Fig. 5l, m and Supplementary Fig. 7t). These results indicate that COPI vesicle-associated TANGO6 serves as a major carrier that directs RPB2 nuclear import to ensure cell cycle progression and proliferation (Supplementary Fig. 7u).

## TANGO6-RPB2 axis is widely used by HSCs during hematopoiesis

Subsequently, we explored the effects of the TANGO6-RPB2 axis in vivo. First, we created conventional *Tango6* knock-out (KO) mice. Unfortunately, no homozygous mutant offspring were identified even at E9.5 (Supplementary Fig. 8a). This is consistent with the observation that *Tango6* is ubiquitously expressed in embryos and highly enriched in the brain from E9.5 to E11.5 (Supplementary Fig. 8b), implying its indispensability in early embryonic development. We generated a *Tango6* conditional knockout allele with loxP sites flanking exon 2 (Supplementary Fig. 8c). We primarily focused on well-established hematopoietic systems in our hand. Immunofluorescence images indicated that TANGO6 co-localized with COPA in both human CD34+ HSCs[50] and mouse bone marrow cells (Fig. 6a, b). We bred *Tango6*^f/f mice with *Vav-iCre* (expressing active Cre in hematopoietic cells[51]) to obtain *Vav-iCre/Tango6*^f/f mice and confirmed Cre-mediated deletion of floxed *Tango6* (Supplementary Fig. 8d–f). Characterization of *Vav-iCre/Tango6*^f/f (cKO) mice at different time points indicated that the cKO mice displayed a pale anemic appearance and died at E15.5–17.5 (Fig. 6c and Supplementary Fig. 8g, h). Because HSCs originate from embryonic aorta-gonad-mesonephero (AGM) via endothelial-to-hematopoietic transition during E9.5 to E12.5[52], we first examined the vascular tissue at E9.5, E12.5, and E14.5. The immunofluorescence staining images indicated that PECAM1+ (vascular endothelial cell marker[53]) signals did not present notable alterations during E9.5–E14.5. However, VE-cadherin+ (vascular endothelial cell marker[54]) signals were comparable at E9.5 but decreased considerably at E12.5 and E14.5 in the cKO mice than WT controls (Supplementary Fig. 8i–l). These data indicate that the blood vessel was only affected to a limited extent at the early stages when HSCs were produced, although their

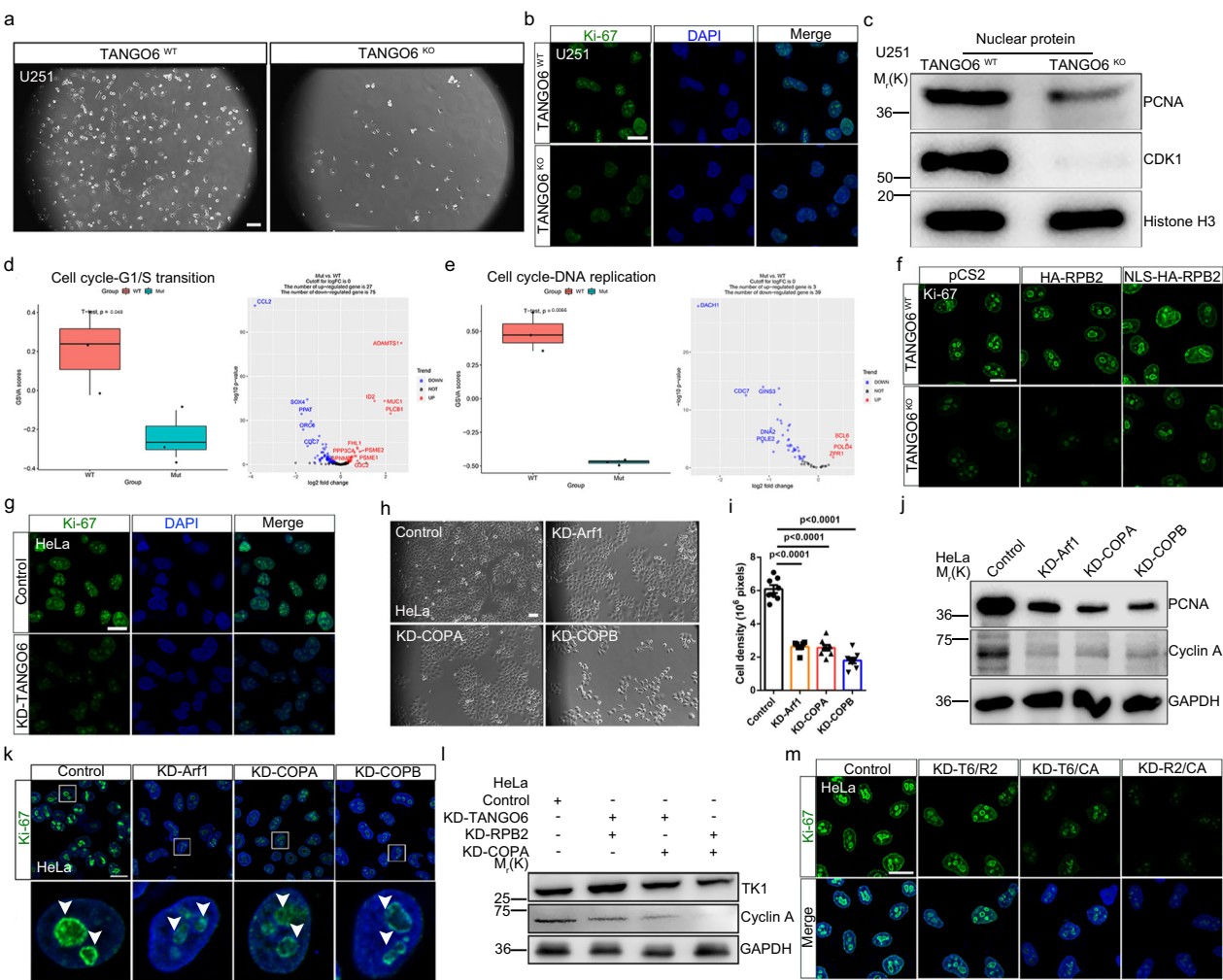

**Fig. 5 | TANGO6 regulates cell proliferation via promoting G1/S transition.**
**a** The cell density of *TANGO6*^WT and *TANGO6*^KO by wide-field microscope. Scale bar, 20 μm. The immunofluorescent staining images of Ki-67 (**b**) and western blot analysis of PCNA and CDK1 in nuclear protein from the *TANGO6*^WT and *TANGO6*^KO cells (**c**). Histone H3 is used as internal standard. Scale bar, 20 μm. GSVA analysis of G1/S transition (**d**) and DNA replication (**e**) process in cell cycle between *TANGO6*^WT and *TANGO6*^KO cells. In the box plot, boxes indicate 25th to 75th percentile, its middle line the median, whiskers the 5th to 95th percentile, and individually plotted data points the outliers. The left panel is the GSVA scores and the right panel is the related genes enriched in corresponding process. **f** The immunofluorescent staining images of Ki-67 after transfecting pCS2 (control), HA-RPB2 and NLS-HA-RPB2 plasmids to *TANGO6*^WT and *TANGO6*^KO cells. Scale bar, 20 μm. **g** The immunofluorescent staining images of Ki-67 after knocking down TANGO6. Scale bar,

20 μm. Wide-field images of cell density (**h**) and corresponding statistics (**i**) when knocking down Arf1, COPA and COPB (**i**, Control, 6.09 ± 0.24; KD-Arf1, 2.62 ± 0.10; KD-COPA, 2.56 ± 0.17; KD-COPB, 1.79 ± 0.17. *n* = 8 visual field in each group). Each point in **i** denotes the cell density in a visual field). Scale bar, 20 μm. Western blot analysis of PCNA and Cyclin A (**j**) and the immunofluorescent staining images of Ki-67 (**k**) when knocking down Arf1, COPA and COPB. The bottom panels are the amplified images of boxed region. The white arrowheads indicate Ki-67⁺ signals. Scale bar, 20 μm. Western blot analysis of TK1 and cyclin A (**l**) or immunofluorescent staining images of Ki-67 (**m**) after double knocking down TANGO6/RPB2 (T6/R2), TANGO6/COPA (T6/CA) and RPB2/ COPA (R2/CA). GAPDH is the internal standards. Scale bar, 20 μm. Statistical significance for **i** was assessed using unpaired one-tailed Student's *t* test. Mean ± s.e.m. Source data are provided as a Source Data file.

development was later affected in the cKO mice. Concordantly, Runx1⁺ signals (a marker of HSCs[55]) comparably appeared in the cKO and WT mice at E9.5 (Supplementary Fig. 8j), indicating an unaffected emergency of HSCs in the cKO mice. *Tango6* transcripts were enriched in the fetal liver at E12.5 (Fig. 6d and Supplementary Fig. 8m) when hematopoietic stem cells (HSCs) expand rapidly[56]. The fetal livers of cKO mice were pallid and smaller than those of the WT sibling controls (Fig. 6e). The amounts of fetal liver cells decreased significantly in the cKO than the siblings (E13.5-E15.5) (Supplementary Fig. 8n). The Albumin⁺ hepatoblasts and c-Kit⁺ and Runx1⁺ HSPCs reduced statistically in the cKO mice at the same time (Supplementary Fig. 8o, p). The LSK (Lin⁻Sca-1⁺c-Kit⁺) fractions (HSCs)[57] were reduced in E15.5 cKO mice (Fig. 6f, g). In the LSK pools, the proportion of SLAM-HSCs decreased obviously in the fetal liver of cKO mice in compared to their WT controls (Fig. 6h, i). The transcriptional levels of *Cyclin B, Cyclin D, Ki-67,*

and *PCNA* decreased substantially in the LSK population of cKO mice compared to those of WT controls (Fig. 6j). Concordantly, the number of Runx1⁺ cells and Ki-67⁺ signals decreased notably in the cKO fetal liver at E15.5 (Supplementary Fig. 8q, r). However, the active-Caspase3⁺ signals were comparable between cKO and WT mice (Supplementary Fig. 8s).

The results of FACS indicated that the Ter119⁺ erythrocytes and CD11b⁺Gr1⁺ granulocytes[58] were profoundly diminished. However, the CD11b⁺Gr1⁻ macrophages[58] in cKO mice were unchanged (Supplementary Fig. 8t). The immunofluorescent staining images indicated that the CD71⁺ and Ter119⁺ (two markers of erythrocytes[59]) signals reduced obviously in cKO mice (Supplementary Fig. 8u). Double staining indicated that RPB2⁺ signals were evenly distributed in WT Runx1⁺ cells but diminished in the nuclei of cKO Runx1⁺ cells (Fig. 6k). qRT-PCR and western blot analysis results indicated that RPB2 activity was

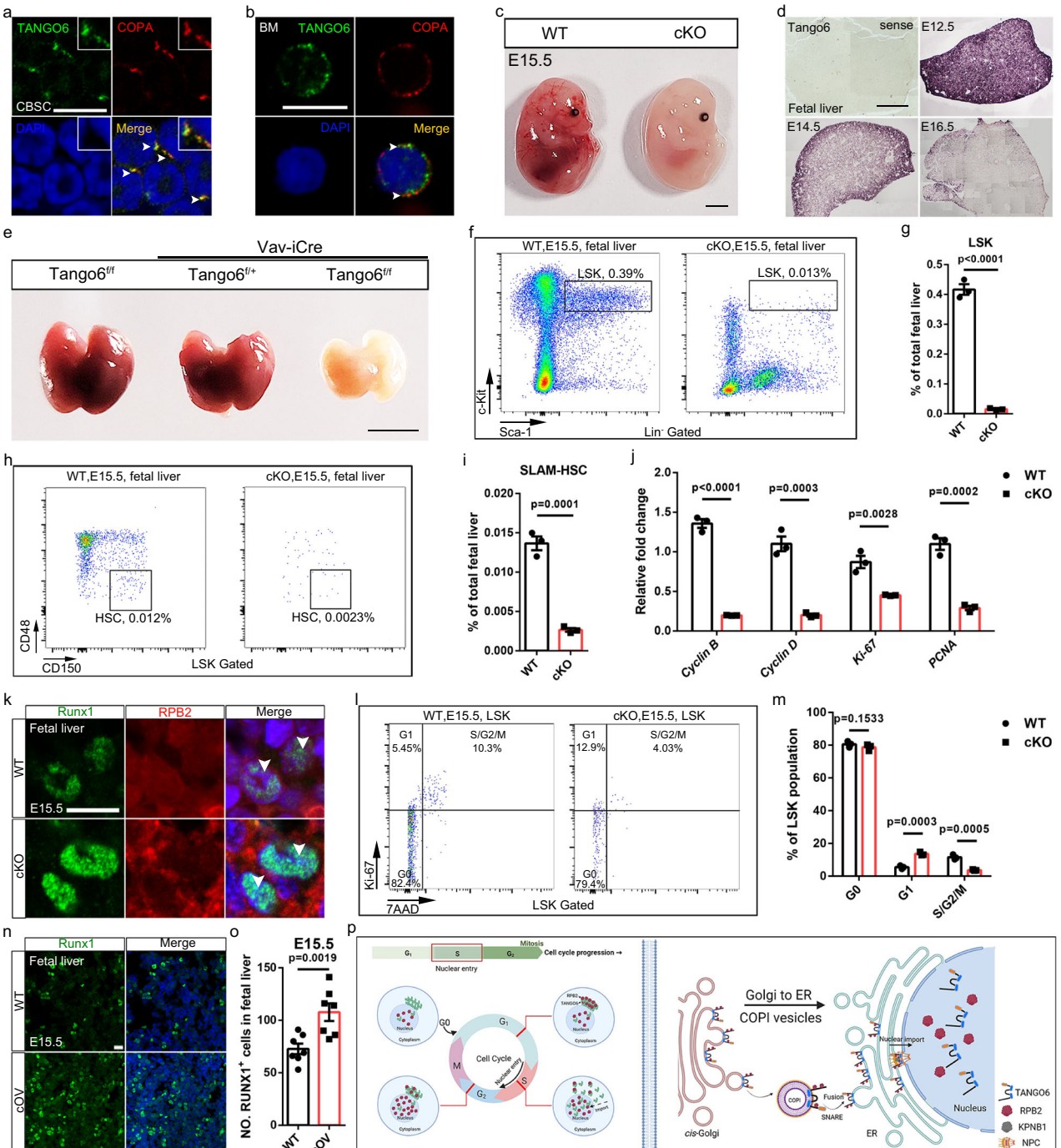

**Fig. 6 | TANGO6-RPB2 axis is critical for mice HSCs expansion in vivo.** The immunofluorescent staining images of TANGO6 and COPA in human cord blood stem cells (CBSC) (**a**) and mice bone marrow cells (BM) (**b**). The white arrowheads indicate co-localized signals. E, Embryonic. cKO, conditional knock out. Scale bar, 20 μm (**a**), 10 μm (**b**). **c** Photographic images of WT and cKO embryos at E15.5. Scale bar, 20 mm. **d** WISH of *Tango6* in E12.5-E16.5 fetal livers. Scale bar, 20 μm. **e** Appearance of the E15.5 fetal livers. Scale bar, 2 mm. **f** Representative FACS profiles of the frequency of LSK (Lin⁻sca-1⁺c-kit⁺) cells in the fetal livers. **g** The frequency of LSK cells within total fetal liver cells at E15.5. **h** Representative FACS profiles of the frequency of SLAM-HSC within fetal liver cells. **i** The statistical graph of proportion of SLAM-HSC cells in E15.5 fetal livers. *n* = 3 mice in each group). **j** qRT-PCR result in the LSK cells from E15.5 fetal livers. *n* = 3. Each dot represents an independent experiment. **k** The immunofluorescent staining images of Runx1 and

RPB2. Scale bar, 10 μm. **l** Cell-cycle analysis of E15.5 WT and cKO in the LSK cells. **m** The statistical graph of the cell-cycle distributions in LSK cells. *n* = 3 mice in each group. The immunofluorescent staining images of Runx1 in WT and cOV (conditional overexpression) mice (**n**) and corresponding statistical graph (**o**). (*n* = 7 visual field in each group). Each point in **o** denotes the number of Runx1⁺ cells in a visual field. Scale bar, 20 μm. **p** Our working model. TANGO6 and RPB2 distribution during various interphases of cell cycle and their interaction mainly occurs at G1/S phase to assure RPB2 nuclear import (left). TANGO6 is anchored to COPI vesicle (purple) via its TM domains (blue). And its N- and C- fragments expose to the cell cytoplasm to capture RPB2 (dark red). TANGO6 carries RPB2 to the nucleus via assistance of ER and KPNB1 (gray) (right). TANGO6-RPB2 axis is functionally conserved in controlling tumor cells and mice HSCs expansion. The models were created with BioRender.com. Source data are provided as a Source Data file.

significantly lower in the fetal livers of cKO mice than in their WT siblings (Supplementary Fig. 8v, w). We also tested the cell cycle status of LSK cells using Ki-67/7-AAD assay[60], revealing a significantly higher frequency of the G1 phase cells in cKO mice (13.61% ± 0.60%) than in WT mice (5.44% ± 0.57%) (Fig. 6l, m). These data indicate that *Tango6* deficiency compromises RPB2 nuclear entry and cell cycle arrest at the G1/S phase in mouse HSCs, which is consistent with our results in HeLa and U251 cells. To validate this conclusion, we cultured the isolated LSK+ HSCs and observed a significant reduction in the colony numbers of LSK+ cells in the BFA treatment and packaged RPB2 lentivirus groups (Supplementary Fig. 8x). We generated *Vav*$^{iCre/WT}$*Rosa26*$^{Tango6/WT}$ transgenic mice (cOV) to achieve the hematopoiesis-specific over-expression of *Tango6*. Immunofluorescence staining indicated that Runx1+ and Ki-67+ cells were increased in cOV mice compared to the controls (Fig. 6n, o and Supplementary Fig. 8y). Therefore, the TANGO6-RPB2 axis is indispensably conserved during embryonic HSCs proliferation in mice. These results revealed that COPI vesicle-associated TANGO6 carries RPB2 to the ER via retrograde transport and then to the nucleus via the TANGO6 NLS with the assistance of importins. The synergistic effects of TANGO6, RPB2, and COPI vesicle transport systems are tightly orchestrated to ensure G1 to S phase progression during cell proliferation, which is essential for the maintenance and expansion of cells in multiple biological processes, including tumorigenesis and hematopoiesis (Fig. 6p).

## Discussion

Previous studies reported that Pol II complex, composed of 12 subunits, should be fully assembled before being imported into the nucleus[6]. However, the nuclear import of large and small subunits depends on distinct mechanisms was also mentioned[61]. Several factors such as GPN-loop GTPase 1/3 (GPN1/3) and RPAP2 have been associated with the nuclear localization of RPB1[9,62]. IWR1 directs the nuclear import of RPB1 in yeast[8]; however, whether the complex is assembled in the cytoplasm or nucleus remains controversial[61]. A possible explanation is that large subunits are carried by specific factors but small subunits engage in passive diffusion. Although the mechanisms utilized by RPB1 are probably adopted by other subunits, whether the large subunits employ special factors for their nuclear import is unclear. TANGO6 deficiency impairs RPB2 nuclear import. But the nuclear entry of other 11 RNA polymerase II (Pol II) subunits was not strongly affected. The interaction of TANGO6 with RPB2, but not with RPB1, in the G1/S phase was critical for its nuclear entry. This phenomenon implies the utilization of specific trafficking machinery for individual subunits.

Immunofluorescence images revealed that TANGO6 was localized in both the cytoplasm and nucleus of multiple mouse and human cells, including U251, HeLa, and HSCs. In the cytoplasm, TANGO6 anchors to the COPI vesicle membrane via a signal anchor and two TM elements. However, both its N- and C-termini, including the signal anchor, are located in the cytoplasm. Two types of transmembrane proteins—integral and peripheral membrane proteins—were observed[63]. The integral membrane proteins span the entire membrane, while the peripheral membrane proteins are associated with the membrane via electrostatic/hydrophobic interactions or hydrophobic segments. TANGO6 is a peripheral membrane protein that contains two TMs and superficially immerses in membranes. This topology helps it separate from the membrane properly.

An important function for the signal anchor is promoting the clathrin-coated vesicle mediated secretion[64]. However, the signal anchor of TANGO6 could not be cleaved, similar to that of MP19, a non-secretory protein mainly localized at the cell membrane[65]. RNA-seq data indicated the limited alteration of the secretory pathway between the mutant and WT cells. Quite limited alterations of secretory proteins, including EGF and Collagen I, were observed in the cell lysate and supernatants upon TANGO6 knockdown in HeLa cells. Interestingly,

the pulse-chase assay suggested an even accelerated secretory process in the *TANGO6*$^{KO}$ cells. Whether this phenotype is caused by *TANGO6* or an indicator or outcome of compromised proliferative cells deserve future exploration. Most secretory protein maturation and signal peptides cleaved at ER[66]. But TANGO6 was not a secretory protein and probably matured in the Golgi. Cargo recognition by COPI is largely mediated by sorting signals in the cytoplasmic domains of the retrieved proteins, which are then sorted into COPI-coated vesicles[67]. Considering the interaction affinity of the N- and C-termini of TANGO6 with RPB2, we presume that the mechanism of TANGO6 on COPI vesicles probably optimizes its interaction with RPB2. This unusual approach utilized by TANGO6 to carry RPB2 onto COPI vesicles revealed an interesting mechanism that provides insight into the role of the vesicle transport system in cell cycle regulation. However, there existed the non-colocalized signals of TANGO6 and RPB2. And a large number of additional factors interacted with TANGO6 in the MS data, pointing to extended functions of TANGO6 in the future studies.

The dynamics of the Golgi complex were mainly investigated in the mitotic phases of cell division[68]. From the late G2 phase to the telophase, the Golgi ribbon undergoes continuous and invertible steps to ensure accurate partitioning and inheritance by daughter cells. However, the signatures of Golgi complex in the interphase of the cell cycle were unremarkable. Interestingly, we observed dynamic disassembly of the *mid/cis*-Golgi, while *trans*-Golgi was only altered to a limited extent across the interphase. The *cis-Golgi* mainly communicated with ER via COPI vesicles; however, *trans*-Golgi was intensively involved in producing secretory vesicles. The differing behaviors of various Golgi complex constituents implied distinct trafficking mechanisms in cell proliferation.

The co-localization of the TANGO6-RPB2 interaction with Calnexin alluded to the crucial role of the ER membrane in its nuclear entry. Recent studies have proposed that several factors enter the nucleus through the ER membrane. For example, endocytosed fibroblast growth factor 1 quickly moves to the ER, where LRRC59, an ER membrane-tethered factor, facilitates its nuclear transport through nuclear pores via importins[40]. Atlastin, a membrane-bound ER GTPase, plays a vital role in sustaining the efficient targeting of membrane proteins to the inner nuclear membrane (INM) by preserving ER topology[69]. We suspect that TANGO6 carries RPB2 to the ER membrane via Golgi-to-ER retrograde transport regulated by SNARE complex-mediated membrane fusion, and then moves to the nucleus through a similar ER membrane-mediated pathway.

TANGO6 deficiency induces cell cycle arrest at the G1 phase, whether in U251 cells or mouse HSCs. The strongest interaction between TANGO6 and RPB2 occurred at the G1/S phase. The levels of RPB2 increased from the G1/S phase and were considerably enhanced in the S and G2 phases, particularly in the nucleus. These data imply that TANGO6 captures RPB2 to the nucleus largely at the G1/S phase. The loss of TANGO6 blocks RPB2 nuclear import, which arrests cell cycle at the G1 phase. This mechanism indicates the unrealized trafficking of COPI vesicles in harnessing the nuclear entry of specific factors during cell proliferation. However, whether this approach is specifically applied by TANGO6 and its binding proteins, such as RPB2 and de novo in the cytoplasm, has not been confirmed. Moreover, whether this method can be efficiently applied by intensively proliferating cells in the context of drastic tumor expansion and early embryonic development is unclear.

Aberrant nuclear enrichment of RPB2 is observed in Runx1+ cells after *Vav-iCre*-mediated hematopoietic deletion of *Tango6*. These data verify the conserved function of the TANGO6-RPB2 axis in the expansion of embryonic HSCs in mice. HSCs stem from the vascular system of AGM at the embryonic stages[52]. Assessment results revealed that the vascular tissues were not strongly affected at the embryonic stages when HSCs were produced, although the VE-cadherin+ signals were reduced from E12.5 in the cKO mice. A recent study indicated that

the hypoxia condition reduced VE-cadherin levels, implying the possible occurrence of hypoxia in cKO mice owing to the limited oxygen supplement by the compromised erythrocyte production[70]. We suspected that the vascular structures were initially generated, but the VE-cadherin levels decreased significantly at E12.5-14.5 in cKO mice by the hypoxia. At the same time, the hepatocytes, including hepatoblasts, were affected in the cKO mice. These probably led to the small and pale appearance of fetal liver in the cKO embryos. However, we could not overlook the possible influence on the blood vessel and fetal liver development in the cKO mice owing to the involvement of Vav-iCre in endothelial cells. Interestingly, the cKO mice harbored a population of c-kit⁻Sca-1⁺ cells, which probably expressed high levels of Sca-1 under a special condition, like hypoxia, in the fetal liver of cKO mice with limited blood. Because a previous study reported that hypoxia led to a significant increment of Sca-1 in the mesenchymal stem cells[71]. We will explore the signatures of Sca-1⁺ cells in cKO mice in the future. We observed early embryonic lethality in conventional *Tango6* KO mice, indicating the vital role played by the TANGO6-RPB2 axis in survival. Consistently, U251 and HeLa cells displayed delayed growth when TANGO6 expression was suppressed. This indicated that the regulatory function of the TANGO6-RPB2 axis is conserved across multiple cellular systems. TANGO6 is a promising target for the efficient expansion of various stem and progenitor cells or for the arrest of specific neoplastic cells, which warrants further investigation.

## Methods

### Study approval

The cord blood stem cells in this study were collected from the healthy parturient. All the donors or their guardians provided written informed consent for sample collection and data analysis. We obtained consent from the donors to publish identifying information. All procedures were approved by the Ethics Committee of the Institute of Chongqing Medical University (Chongqing, China).

### Mice

*Tango6* conventional knock out (KO), conditional knock out (cKO) and conditional overexpression (cOV) mice (see the Supplementary Table 4) were generated in C57BL/6 strain by Cyagen Biosciences. The conventional KO mice were produced by the deletion of 6661 bp between exon 2 and exon 3 of *Tango6* gene. The conditional KO (*Tango6*^(f/f)^) mice were constructed by homologous recombination using the targeting vector that contains loxP sites flanking exon 2 of the *Tango6* locus. The conditional overexpression (cOV) mice were generated by the knock-in method at Rosa 26 locus, which was a ubiquitous locus that permitted an exogenous strong promoter inserted within it to drive high expression[72]. The donor vector containing "CAG promoter-loxP-PGK-Neo-6*SV40 pAloxP-Kozak-Mouse Tango6 CDS-P2A-EGFP-rBG pA" cassette together with gRNA and Cas9 mRNA were co-injected into the fertilized mouse eggs to generate targeted conditional knock-in offspring. These mice were mated with *Vav-iCre* alleles (given by the Southwest Hospital of Army Military Medical University) to generate *Vav-iCre/Tango6*^(f/f)^ mice (cKO) and *Vav-iCre/Rosa26*^(Tango6/WT)^ mice (cOV). The mice were identified by specific primers (See the Supplementary Table 1). The maintenance and experiment procedures of mice were complied with the guidelines approved by the Institutional Review Board of Southwest University (Chongqing, China). This guidance ensured a clean and disease-free comfortable living environment for the animals under a 12:12 h light−dark photoperiod (21 ± 2 °C).

### Cell lines

The U251, HeLa and 293T cell lines (See the Supplementary Table 4) were cultured following a standard protocol[73]. In brief, the cells were cultured in the medium contained high glucose DMEM with glutamine and sodium pyruvate (Gibco), 10% fetal bovine serum (Biological industries), 1% L-Glutamine (Beyotime), 1 μg/mL Mycoplasma Removal Agent Plus (Beyotime), 1% Penicillin-Streptomycin-Amphotericin B Solution (Beyotime) and incubated in the cell-incubator (37 °C, 5% $CO_2$) (Shanghai Yiheng Technology). In order to generate the *TANGO6* mutants in the U251 and HeLa cells, the guide RNAs ("gRNA1: 5′-ATGGCGGCCCGACAGGCCGT-3′, gRNA2:5′-ATGCGGTCTGGATCGGATTT-3′, gRNA3: 5′-GCCTGTCGGGCCGCCATGAC-3′") were designed. The strategies were conducted by the Genloci Biotechnology. The *TANGO6* mutant cells were identified by sequencing. The U251 homozygous mutant cells (*TANGO6* ^KO^) presented heterozygosity of simultaneous deletion of 41 and 86 bp in each strand of TANGO6.

### siRNA library screening

A siRNA library of human membrane traffic gene was purchased from the company (RiboBio, China). We conducted a screening experiment following a standard protocol[74]. In brief, we mixed 3 pairs of siRNA for each gene in equal proportion and delivered them to the U251 cells. The cells ($0.5–2 × 10^5$) were pre-plated in 12- or 24-well plates for 24 h. The transfection was carried out using Lipo8000 transfection reagent (Beyotime, China) according to the manufacturer's instruction. The final concentration of siRNA was 40 pmol (24 well) or 60 pmol (12 well), respectively. After 48 h, the cell number was counted by hemocytometer (Marienfeld, Germany).

### Plasmid transfection

U251 and HeLa cells were pre-plated in 6- or 24- well plates for one day and then seeded to 70−80% confluent prior to transfection. Transfection was carried out using Lipo8000 transfection reagent (Beyotime, China) according to the manufacturer's instruction. In brief, 1 μg plasmid and 3 μL Lipo8000 was mixed and diluted by DMEM (Gibco). After incubating for 15 min, the mixture was added to culture the samples for 48 h to promote the proteins production.

### Knocking down assays

The siRNA duplexes targeting human TANGO6, COPA, COPB, Arf1, KPNB1, RPB2, LRRC59, AP1M1 and their nontargeting controls (See the Supplementary Table 3) were purchased from Ribobio. The siRNA transfection assay was performed following the protocol of the LipoRNAi™ Transfection Reagent (Beyotime). The knocking-down efficiencies were verified by western blot analysis.

### Staining assays

The RNA probes were in vitro synthesized by using the Digoxin (DIG) RNA Labeling kit T3/T7/SP6 (Roche) according to its manufacturer's instruction. The sagittal frozen section of mouse fetal livers at different stages were performed by using a Leica CM1850 machine following per standard protocol[75]. The in situ hybridization of mice embryos and frozen sections were performed following a standard protocol[76]. The pictures were captured by a Carl Zeiss Discovery V20 microscope. To conduct the immunofluorescent staining on the slices, the tissue pieces were rinsed in PBDT (1× PBS with 1% BSA, 1% DMSO, 0.5% Triton X-100) and blocked with buffer (2% FBS and 0.1% Tween 20 in 1 × PBS) at room temperature (RT) for 1 h. Primary antibodies were utilized to culture the slices at 4 °C overnight. Afterwards, the samples were stained with Alexa 488/555/647-conjugated secondary antibodies (See the Supplementary Table 2) at 37 °C for 3 h. Immunofluorescent staining for HeLa or U251 cells was performed according to standard protocols[77]. In short, the cells on cover slips were fixed with 4% PFA at RT for 10 min and then treated with 0.2% Triton X-100 at RT for another 15 min. Then, they were blocked with phosphate-buffered saline (PBS) containing 5% fetal bovine serum (FBS) at RT for 1 h and incubated with primary and secondary antibodies sequentially (Primary antibody, 4 °C, overnight; secondary antibodies, RT, 2 h). Immunofluorescent staining images were taken by either Zeiss LSM700 or LSM880 with Airyscan confocal microscopes.

## Super-resolution images

The incubations of primary and secondary antibody (See the Supplementary Table 2) were firstly conducted following a standard protocol, according to manufacturer's instruction of STED (Stimulated Emission Depletion) microscopy. The autofit z-axis was chosen based on the position of target signals in the cells on the cover slips. Then the proper images were captured by laser beam scan on the corresponding x-axis and y-axis. The STED images' deconvolution was processed by the Huygens Professional (Scientific Volume Imaging) to improve contrast ratio and SNR (SIGNAL-NOISE RATIO). The images of transmission electron microscope were conducted by Wuhan Servicebio technology CO. We prepared the sample according to their protocols[78].

## Photo-conversion

We conducted the Photo-conversion experiments as that an established protocol[36]. In brief, the 63x oil objective in a Zeiss LSM880 confocal microscope was used. The area of cytoplasm was photo-converted by a 405 nm laser with 10% output, which partially converted TANGO6-mMaple3 from the original green to red color. After that, the process of red signals entering the cell nucleus was tracked and analyzed by the ZEN software.

## FACS

Single-cell suspensions of mice fetal liver were fulfilled as that per standard protocol[79]. In brief, fetal liver were cut into pieces and then digested with collagenase IV (Sangon Biotech) at 37 °C for 30–60 min. Single-cell suspensions were obtained by filtration of 70 μm cell strainer. Erythrocytes were lysed using red blood cell lysis buffer (Beyotime) and then incubated with the corresponding antibodies (See the Supplementary Table 2) in the dark at 4 °C for 1 h. The flow cytometry analyses, including LSK, SLAM-HSCs and cell cycle, were performed using the Cytek® Northern Lights. The flow cytometry sorting and analyses were performed using the MoFlo XDP (Beckman Coulter). All the experiments followed the manufacturer's instruction.

## Isolation of hematopoietic stem cells from human umbilical cord blood

Human cord blood samples were collected from The Third Affiliated Hospital of CQMU (Chongqing, China) after obtaining informed consent with the compliance of the institutional review board. We isolated CD34+ hematopoietic stem/progenitor cells from umbilical cord blood as an established protocol[50,80]. In brief, we firstly isolated mononuclear cells following the ficoll-paque density gradient method (density 1.077 g/mL, Sigma Aldrich). Then, the CD34+ cells were isolated by using magnetic beads on an affinity column according to the manufacturer's instructions (Miltenyi Biotec, Bergisch-Gladbach, Germany). Ultimately, we added fresh CD34+ cells on coverslips and performed the immunofluorescent staining accordingly.

## Lentivirus package and mouse HSCs in vitro culture

The lentivirus package was performed according to a standard protocol[81]. In brief, $5 \times 10^6$ 293T cells were put on 10 cm dish and co-transfected with 10 μg pLKO. 1 shRNA, 7.5 μg of psPAX2 and 3 μg pMD2.G plasmids to produce lentivirus particles. The supernatant containing viral particles was harvested twice at 48 h and 72 h after transfection, which was then filtered through Millex-GP Filter Unit (0.22 mm pore size, Millipore). Then, the Lenti-Concentration Virus Precipitation Solution was added (ExCell Bio) and incubated in 4 °C for 24 h. After that, centrifuge ($600 \times g$, 15 min, 4 °C) was performed. The collected viral particles were resuspended in PBS containing 0.1% BSA, and stored at −80 °C. To culture the mice HSCs in vitro, the LSK cells from adult mice (3 month) bone marrow were sorted via FACS. These cells were cultured in the medium (Procell, CM-M309) and incubated in the cell-incubator (37 °C, 5% CO₂) (Shanghai Yiheng Technology) for 24 h. Then, the cells were treated with BFA (2 μM, 24 h) or packaged

RPB2 lentivirus (20 μL, 48 h) transfection. The colony number was eventually counted under the wield-field microscope.

## Cell cycle synchronization and cell cycle analysis

The double thymidine block assay was conducted to induce cell cycle synchronization according to a standard protocol[20]. In short, the HeLa cells were put on the coverslips at a 24-well cell culture plate. After incubation for 24 h, we added 2 mM thymidine (Selleck) to culture these cells for an additional 16 h. The cells were then washed using 1× PBS three times. And they were supplied with fresh cell culture medium to incubate for another 12 h. Then, we added 2 mM thymidine again and incubated them for 16 h to collect the cells at G1/S phase. Subsequently, we re-incubated the cells with fresh medium for 4 h to obtain the cells at S phase. We sequentially collected the cells of G2 phase after another 7 h. The cells of G0 phase were gained by serum starvation. To analyze cell cycle of the LSK cells, we conducted Ki-67/7-AAD assay. The details can be found in ref. 60. In brief, the fresh fetal liver cells were stained with antibodies against LSK (Lin⁻Sca-1⁺c-Kit⁺) cells. Then, these cells were stained with Ki-67 and 7-amino-actinomycin D (7-AAD) according to the manufacturer's instructions (Biolegend).

## Pulse-chase assay

To examine the protein secretion abilities, the pulse-chase assay was conducted[35]. Briefly, we added 2 mM stable isotope [2,3-¹³C₂]alanine in the cell medium. After incubation for 12 h, the cells were washed using 1× PBS by three times. And they were supplied with fresh cell culture medium. We collected cell pellet and medium supernatant at different time points (0 h, 2 h and 6 h). These samples were sent to Metabo-Profile Biotechnology (Shanghai) Co., Ltd (See the Supplementary Data 1, pulse-chase assay analysis). The project was performed under the guidance of Quality Management System ISO 9001:2015 (QAIC/CN/170149).

## Vector construction and chemical treatment

To validate the genetic sequences of the NLS (Nuclear localization signal) in TANGO6 gene, we deleted the whole NLS by Seamless Cloning Kit (Beyotime) according to the manufacturer's instruction. To perform site-directed mutagenesis, we precisely omitted or altered the sequence of targeted amino acids on the constructed pCS2-HA-TANGO6 plasmids following a standard protocol[82]. In short, the primers were designed on both sides of the mutation site. The fragments were amplified by PCR and they were used to obtain the plasmids containing site-directed mutagenesis. To verify TM (transmembrane) and signal anchor domains, the original or modified elements were fused with the DsRed fragments by seamless cloning. For chemical treatment assay, the HeLa cells were treated with thapsigargin[39] (2 μM, Selleck) for 24 h to induce ER stress. For topology detection and cell cycle synchronization (see below), digitonin (25 mg/mL, Sigma) or thymidine (2 μM, Selleck) were individually used to treat HeLa cells on ice for 5 min (digitonin) or for 16 h (thymidine).

## Mass spectrometry analysis

According to a standard protocol of Lipo8000™ Transfection Reagent (Beyotime), 2 μg pCS2-FLAG-TANGO6 plasmids were transfected into the U251 cells. They were pulled down by FLAG-beads (Smart Life Sciences). TANGO6 cross linked beads (invitrogen) were used to pull down endogenous proteins. These beads ($n = 1$ in each group) were collected in 100 μL buffer (100 mM NH₄HCO₃, 10 ng/μL Trypsin) and sent to SIMM (Shanghai Institute of Materia Medica, Chinese Academy of Science) to perform mass spectrometry (Orbitrap fusion). The Maxquant (1.6.2.3) software package was used to analyze the data from mass spectra (See the Supplementary Data 2, mass spectrum analysis). Peptide identification was filtered at a false discovery rate (FDR) < 1%.

## RNA sequencing and GSVA analysis

The total RNA of TANGO6 [WT] and TANGO6 [KO] samples were extracted with Trizol reagent (Sangon Biotech) according to the manufacturer's protocol. The purified total RNA samples were sent to Guangzhou Gene Denovo Biotechnology Co. Ltd and cDNA library was separately constructed before sequenced with Illumina HiSeqTM 2000 using the paired-end technology (PE100). Differentially expressed genes (DEGs) between the two clusters (WT vs Mut) were identified using R package "Limma". Gene set variation analysis (GSVA) is used as a non-parametric, unsupervised method to estimate the variation of pre-defined gene sets in case and control samples of gene expression data[47]. All the reference gene sets were derived from the Gene Ontology (GO) Biological Process ontology in Human Molecular Signatures Database (MSigDB, https://www.gsea-msigdb.org/gsea/msigdb/collections.jsp). The GSVA scores were ranging from −1 to 1. A positive GSVA score means that the pre-defined gene set in a sample has a greater expression that the same gene set with a negative value.

## Sucrose density gradients centrifugation

The homogenate from $1 \times 10^7$ HeLa cells were collected by freeze-thawing in liquid nitrogen[45]. The homogenate was mixed with proteinase inhibitor (Roche) and centrifugated at a low speed (1000 g/min, 4 °C, 5 min) to remove cell nucleus. The supernatant was then collected and centrifugated at a medium speed (20,000 g/min, 4 °C, 10 min) to wipe off large organelles. Finally, the isolated liquid supernatant was loaded onto 10% to 37% sucrose gradients to perform centrifugation at a high speed (100,000 g/min, 4 °C, 3 h). Nine equal fractions were eluted from the bottom and lysed by 1× SDS lysis buffer (Beyotime). Each fraction lysate was detected by organelles markers.

## Proteins extraction

1× SDS-PAGE sample loading buffer (Beyotime) was used to extract the total protein according to manufacturer's instruction. The nuclear and cytoplasmic proteins were isolated by a kit (Beyotime) following its manufacturer's instruction. The supernatant proteins of medium were concentrated and extracted as an established protocol[83]. Briefly, we rinsed an ultrafiltration device containing 10 kDa molecular weight cut-off (MWCO) filter (milipore, 15 ml) with PBS. Then, we added cell medium into the filter to perform centrifugation ($1000 \times g$, 30 min, 4 °C). Lastly, we collected the supernatant to conduct western blot. For Golgi and ER extraction, the specific organelles were isolated by using Golgi Apparatus Enrichment Kit (invent) and ER Enrichment Kit (invent), according to the manufacturer's instruction.

## Nuclear extract preparation

The nuclear extract and pellet isolation was performed as a standard protocol[36]. Firstly, we collected $2 \times 10^6$ HeLa cells and washed them by 1× PBS. Then, these cells were resuspended in 3× volume of Buffer A (10 mM Tris-HCl (pH = 7.9), 1.5 mM MgCl$_2$, 10 mM KCl and freshly added 1 mM DTT) on ice for 10 min with 1× cocktail protease inhibitor (Roche) and further lysed by Dounce tissue grinder (~7 strokes). Next, the sample was collected to perform centrifuge ($9700 \times g$, 15 min, 4 °C). The suspension was used as cell cytoplasm and the pellet as rough nucleus. Then the pellet was homogenized in 0.5 volume of Buffer B (20 mM Tris-HCl (pH = 7.9), 25% glycerol, 1.5 mM MgCl$_2$, 0.2 mM EDTA (pH = 8) and 20 mM KCl). The same volume of Buffer C (20 mM Tris-HCl (pH = 7.9), 25% glycerol, 1.5 mM MgCl$_2$, 0.2 mM EDTA (pH = 8) and 1.2 M KCl) was added and the mixture was gently rotated at 4 °C for 30 min. The suspension was isolated as nuclear extract and the pellet was collected as nuclear pellet after centrifugation ($6800 \times g$, 30 min, 4 °C).

## Immunoblotting

We performed immunoblotting as an established protocol[84]. In brief, the protein samples were separated on sodium dodecyl sulfate polyacrylamide gel electrophoresis (SDS-PAGE) gels and then transferred onto polyvinylidene difluoride (PVDF) membranes (Millipore Corporation). The membrane (contained target protein) was blocked in TBST buffer (Beyotime) with 5% nonfat dry milk (2 h, RT) and then incubated with primary antibody (4 °C, overnight). Next, the membrane was washed in TBST several times, and then incubated with secondary antibody (invitrogen). After 2 h, the membranes were washed with TBST several times and were visualized by ECL. The images were taken by an Odyssey two-color infrared fluorescence imaging system (Li-cor, Lincoln, NE, USA).

## Co-immunoprecipitation

The co-immunoprecipitation assay was performed following a standard protocol[85]. Briefly, the plasmids (pCS2-HA-TANGO6, pCS2-FLAG-TANGO6, pCS2-HA-RPB2, pCS2-GFP-N, pCS2-GFP-Lumen and pCS2-GFP-C) and Lipo8000™ Transfection Reagent were transfected into HeLa or U251 cells. After 48 h, the cells were harvested and proteins were purified using cell lysis buffer of western blot analysis and IP (Beyotime). Then, the proteins were mixed with anti-HA/FLAG/GFP beads (Smart Life Sciences) at 4 °C overnight. Finally, the beads were collected to perform western blot analysis according to the manufacturer's instruction or as reported previously.

## Topology detection

TANGO6 topology detection was conducted following the same method as that per standard protocol[16]. In brief, the pCS2-HA-TANGO6-FLAG plasmids were transfected into the HeLa cells and incubated for 15 h. The cells were washed with KHM buffer on ice and then treated with KHM buffer containing 25 mg/mL digitonin (MCE) for 5 min. The cells were washed with KHM buffer once again and then fixed with 4% PFA in 1× PBS at RT for 10 min. The next steps were conducted following the cell staining assays.

## Quantitative RT-PCR

Quantitative RT-PCR (qRT-PCR) was conducted according to a previous method[84]. Mice LSK$^+$ cells were sorted by flow cytometry (Moflo XDP, Beckman). REPLI-g WTA Single Cell Kit was used to generate cDNA libraries following its procedures. All the data were obtained by using the Eppendorf and Roche qRT-PCR machines and the expression level of β-actin was used as the reference control. The information regarding qRT-PCR primers was listed in the Supplementary Table 1.

## Statistics and reproducibility

The positive signals were manually scored and double confirmed blindly. Statistical analyses were performed by GraphPad Prism6.0. The total and average areas of GM130$^+$ signals were quantified by imaris 9.0.1 according to a previous study[86]. All quantified data (mean ± s.e.m) were analyzed by one-sided Student's $t$ test. All error bars represented mean ± SEM. All "n" and "P" values and statistical tests were indicated in figures and corresponding legends. Probability value $p < 0.05$ was considered statistically significant. The experiments in this study, including immunofluorescent and drugs treatment, WB and qRT-PCR, FACS, were independently repeated at least three times. Similar results were obtained.

## Reporting summary

Further information on research design is available in the Nature Portfolio Reporting Summary linked to this article.

# Data availability

The RNA-seq data had been deposited in the Sequence Read Archive under accession code PRJNA1065112. The mass spectrometry proteomics data have been deposited to the ProteomeXchange Consortium via the iProX partner repository with the dataset identifier PXD049158. All data that support the findings of this study, including

experimental animals, antibodies, vector constructions, and chemicals, are available within the article, its Supplementary Information, or from the corresponding author upon reasonable request. Source data are provided with this paper.

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

## Acknowledgements

We thank W. Pan, H. Chen and Y. Guo for mass spectrometry analysis and discussions, B. Zhou, J. Zhou, H. Huang and D. Wang for FACSs

technical assistances. This work was supported by National Key Research and Development Project (2019YFA802703, 2023YFA1800100), National Natural Science Foundation of China Grants (32270873, 31822033) and Chongqing Population and Health Special Funding of China (CSTB2023TIAD-KPX0056). The figure models were created with BioRender.com.

## Author contributions

Z.F., S.L., L.Li, and L.Luo designed the experiments. Z.F., S.L., M.S., C.S. and C.L. performed most experiments. S.L. and H.Z. conducted mice genotype identification and FACS. X.Z. isolated CD34+ cells from human umbilical cord blood. Y.L. did the RNA sequencing analysis. Y.Z., Y.H., C.R., P.Y., C.W., Y.J., H.L., M.M., F.Z., and L.Luo provided technical assistance. Z.F., F.Z., and L.Li wrote and improved the manuscript.

## Competing interests

The authors declare no competing interests.
