## [Peer Review File · Nature Communications]

TANGO6 regulates cell proliferation via COPI vesicle-mediated RPB2 nuclear entryREVIEWER COMMENTS

Reviewer #1 (Remarks to the Author):

This article by Feng et al provides evidence that COPI vesicle-associated TANGO6 is a regulator of cell cycle progression through the nuclear transfer of the second largest RNA pol II subunit RPB2. The results show that TANGO6 is present in COPI, but not COPII, vesicles and that this interaction is needed for its distribution. In Fig 1, Green and yellow letters on white background not easy to read. Please use different colour settings if possible. The domains required for the interaction have been determined. This is elegantly represented in the figures. The data then went to show that the COPI-SNARE-ER pathway is involved in the nuclear entry of TANGO6. I noted that not all the abbreviations are defined, for example I have not found NSF. Please check all along the text to ensure that all abbreviations are defined, which is especially important to non-expert readers. Then the data shows that the karyopherin subunit beta-1 (KPNB1) participate to TANGO6 nuclear entry. The authors then present data convincingly showing that TANGO6 transports the RPB2 subunit of Pol II in the nucleus. Details of this interaction are provided. Please explain more clearly what happen to the other eleven Pol II subunits. The in vitro data is consistent and globally support the conclusions. The authors then present evidence that COPI vesicle-associated TANGO6 involved in RPB2 nuclear import regulates cell cycle progression and proliferation. The final section of the paper shows that the TANGO6-RPB2-COPI transport system synergises to orchestrate the G1 to S phase progression of the cell cycle, which has an impact in vivo. The Methods, Discussion, References sections are complete and clear. Figures are generally well presented.

Reviewer #2 (Remarks to the Author):

This manuscript by Feng et al. presents a study of the function of Tango6, a protein that has not been studied in detail before. The authors report that Tango6 is a transmembrane protein that localizes in the Golgi, is transported from the Golgi to the ER in COPI vesicles through interaction with COPI, interacts with NSF to undergo SNARE-mediated fusion with the nuclear ER, then strikingly jumps to solubility to enter the nucleus through the nuclear

pore complex and becomes a nuclear protein. All the way, the authors claim, Tango6 is bound to RNA polymerase II subunit B (RPB2), and required to transport RPB2 into the nucleus. The transport of RNA polymerase II B by Tango6 would regulate the cell cycle, coupling a novel S-phase Golgi disassembly (that no one had seen before to my knowledge) to the entrance of RNA Polymerase II B, also a novel mechanism for regulation of cell cycle and G1/S transition.

There are many extraordinary claims here and one can get lost navigating the extended data to figure out what is possible, dubious, unsupported or outright impossible. The fact is that Tango6 localization is highly inconsistent among figures, the topology proposed for the protein is very dubious, basic cell biological mechanisms proposed (spectacularly novel in some cases) are presented with a confidence that is not commensurate with the strength of the evidence presented. Due to these and other many issues, I cannot recommend publication.

Main concerns:

1. How authors decide to study Tango6 needs more explanation. Line 87: "We screened a panel of molecules and identified Tango6... that displayed an identical cytoplasmic dynamic localization as COPA across the cell cycle" does not explain. The screening aim, candidates, approach and controls should be clearly presented.

2. The authors state that Tango6 belongs to the TANGO family, but no such family exists. Tango designates a phenotypic class of genes showing defects in secretion upon knock down in Drosophila S2 cells (Bard et al.). Tango proteins are not evolutionarily or structurally related to each other. The direct nature of their effect on secretion is in many cases quite dubious: Tango 4 is spliceosome Prp19, Tango7 is translation factor eIF3m, Tango9 is a plasma membrane sugar transporter, Tango12 is a sodium pump. Is there a defect in secretion associated with Tango6 loss here?

3. The authors report a dispersion of Golgi COPI and cis-Golgi GM130 in S-phase that (to my knowledge) no one observed before. This should be carefully characterized, quantified and

supported by cell cycle markers, as synchronization does not guarantee the specific cell shown is in a given phase of the cell cycle. In fact, DNA content from DAPI signal in images provided does not agree with S and G2 phase designations.

4. Is this a whole Golgi disassembly or specific to COPI/GM130? The authors seem to suggest with trans-Golgi network marker TGN46 that this is only COPI and GM130, but can they show in the same image that COPI or GM130 are dispersed while the structure of the Golgi remains unperturbed (TGN as well, but better with more integral cis, mid and trans markers).

5. The authors claim that Phobius and UNiprot predict six transmembrane domains and one signal peptide in Tango6. However, I only saw one possible transmembrane domain and no signal peptide in these predictions. Could the authors clarify this?

6. The topology of the protein is in fact extremely unclear. If Tango6 has a signal peptide, how come all three constructs in Fig 1f show the same localization, including nuclear puncta in all three? Signal peptide-DsRed should be just secretable DsRed (after signal peptide cleavage). And how come both N and C termini end up cytoplasmic? Shouldn't HA be cleaved from HA-Tango6-FLAG?

7. The study relies heavily in antibody stainings and subcellular localization studies. However, the localization of Tango6 is highly inconsistent among images. Tango6 antibody, HA-Tango6, HA-Tango6-Flag and Tango6-mMaple3 displayed completely different localization pattern. For example: compare anti-Tango6 in Fig 1a versus Fig S2f. Also HA-Tango6 in Fig2k, Tango6-mMaple3 in Fig S4b. One has to wonder whether Tango6 signals in the nucleus are real or typical antibody staining unspecific specks, or belong to a plane that's not inside the nucleus. Simultaneous anti-Tango6 (green) /anti-FLAG or HA (red) in Fig. 1g does not show good colocalization when checked in detail. Importantly, when stained together in Fig 2n, Tango6 and RPB2 show very little colocalization.

8. If Tango6 has 6 hydrophobic transmembrane regions, how does it come off membrane, passes the nuclear pore and just exists floating inside the nucleus? This cannot be simply

stated without clear proof.

9. This study wants to demonstrate that Tango6 affects cell cycle by mediating RPB2 nuclear transport. Missing experiments: RPB2 deficiency phenotype on cell proliferation; RPB2 supplement rescues Tango6 mutant.

Other points:

1. The ratio of COPI colocalizing with scattered Tango6 is obviously much lower than 60% in Fig1c.

2. Line215. The authors conducted FLAG-Tango6 pull down-MS experiments and identified RPB2. First, controls are missing here. Second: in previous results, Tango6 showed strong interactions with COPI and NSF, which are missing in MS results.

3. The authors claim nuclear import of Tango6 protein in Thapsigargin treatment (2f) and KPNB1 KD (2n) goes down, but overall signal levels are also very reduced. Furthermore, there wasn't any nuclear Tango6 signal in Fig. 1a (see major point 8).

4. The authors omit from their analysis the M phase of the cell cycle, which is when the nuclear membrane breaks down and the Golgi really disassembles. What happens with RPB2 then?

5. Line293. In the heatmap of Tango6 KO vs WT (Fig S6C), the trend of RPB2/3/4/8/9/10 reduction is not clear, and WT-1 is so different compared to others. Furthermore, this result is inconsistent with results shown in Fig S5k and Fig S5n, where RPB2 level is unchanged with Tango6KD.

6. In FigS5k and FigS5m, Tango6 KD show different effect on RPB2 in U251 cells.

7. In Tango6 cKO mice, RPB2 protein level decreases. However, if Tango6 mediates RPB2 import into nucleus, without Tango6 the RPB2 amount should be unchanged, while showing aggregation outside nucleus, like shown in HeLa cell (Fig 3c, 3d, 3i). Is the expression/protein

level of RPB2 affected or not by Tango6 deficiency?

8. Line360. The authors conclude that Tango6, RPB2 and COPI have synergistic effect on G1/S progression. More experiments (such as double KD) are required to say they have synergistic effect on G1/S progression.

9. The target of Tango6 identified here is RNA polymerase II. So why cells with Tango6 deficiency arrest at G1/S phase, instead of any other phase?

10. Figures 4 and 5, presenting loss of function of Tango6, show cell proliferation defects in conditional KO mice, but this is not a terribly specific phenotype, it is compatible with animals and cells just being sort of sick, and therefore the results do not add much support to the proposed Tango6/COPI/SNARE/NPC/RNA Pol II/cell cycle axis.

Reviewer #3 (Remarks to the Author):

The work has high significance with potential identification of intra-organelle transport system for critical cell cycle machinery such as RPB2. The conclusions are supported by the in vitro results.

However, the analysis focusing on the fetal liver expansion of HSCs and the role that TANGO6 plays in this specific aspect of hematopoietic development needs to be further and more carefully investigated. Specifically the flow cytometry analysis of the fetal liver must be conducted using LSK and SLAM markers (CD150+CD48-) as the use of only LSK progenitors markers can be confounded by primitive waves (yolk sac) of developmental hematopoiesis which are not affected by the vavCre driver. Furthermore, while it is commendable that the authors attempted a tissue specific deletion of TANGO6 after the constitutive mutant was embryonic lethal at a very early stage, the concern is that that even the vaviCre mutant appears to have a dramatic phenotype that precludes genuine hematopoietic analysis in the fetal liver. The defect observed in Fig.5c-e is striking and indicates a complete loss of hematopoietic cellularity, so since VavCre is predicted to be active in the hematopoietic compartment starting at ~E12 (Joseph C et al Cell Stem Cell

2013) but can also affect endothelial cells. This would explain the severe pale embryos of the TANGO6 conditional knockouts. Thus the authors must also analyze the emerging HSCs in the aorta gonad mesonephros (AGM) of their mutant mice in order to determine if loss of TANGO6 affects the formation of the HSCs or just their proliferation. Assessment of the vascular tissue at E9.5, E12.5 and E14.5 by staining for VeCadherin and PECAM1 is also required to rule out any defect in vascular development which may preclude any analysis of the fetal liver hematopoiesis. Since TANGO6 is predicted to affect proliferation, angiogenic developing vasculature may also be drastically affected and in fact the pale vaviCre⁺ embryo in Fig.5c suggests a defect during earlier development than the fetal liver. If this is the case, apoptosis markers should also be assessed along with the proliferation analysis, since a defective vascular tissue will negatively impact hematopoietic viability.

Reviewer #1 (Remarks to the Author):

This article by Feng et al provides evidence that COPI vesicle-associated TANGO6 is a regulator of cell cycle progression through the nuclear transfer of the second largest RNA pol II subunit RPB2. The results show that TANGO6 is present in COPI, but not COPII, vesicles and that this interaction is needed for its distribution. In Fig 1, Green and yellow letters on white background not easy to read. Please use different colour settings if possible. The domains required for the interaction have been determined. This is elegantly represented in the figures. The data then went to show that the COPI-SNARE-ER pathway is involved in the nuclear entry of TANGO6. I noted that not all the abbreviations are defined, for example I have not found NSF. Please check all along the text to ensure that all abbreviations are defined, which is especially important to non-expert readers. Then the data shows that the karyopherin subunit beta-1 (KPNB1) participate to TANGO6 nuclear entry. The authors then present data convincingly showing that TANGO6 transports the RPB2 subunit of Pol II in the nucleus. Details of this interaction are provided. Please explain more clearly what happen to the other eleven Pol II subunits. The in vitro data is consistent and globally support the conclusions. The authors then present evidence that COPI vesicle-associated TANGO6 involved in RPB2 nuclear import regulates cell cycle progression and proliferation. The final section of the paper shows that the TANGO6-RPB2-COPI transport system synergises to orchestrate the G1 to S phase progression of the cell cycle, which has an impact in vivo. The Methods, Discussion, References sections are complete and clear. Figures are generally well presented.

We deeply appreciate your great efforts in reviewing our manuscripts. The comments and suggestions are quite useful for our manuscript improvement. We examined other eleven Pol II subunits accordingly. The figures and descriptions were corrected and polished in the revised manuscript. The detailed responses are listed point-by-point below.

1) In Fig 1, Green and yellow letters on white background not easy to read. Please use different colour settings if possible.

Answer: We are sorry for the confused colors and background. The green and yellow letters were changed to atrovirens and black colors accordingly. Thanks!

2) I noted that not all the abbreviations are defined, for example I have not found NSF. Please check all along the text to ensure that all abbreviations are defined, which is especially important to non-expert readers.

Answer: All the abbreviations, including HSC, LRRC59, SNARE, NSF, GPN1/3, RPAP2 Arf1, IWR1, IGF1 and CO-IP, were defined in our revised manuscript (marked by gray background). Thanks very much for your kind suggestions!

3) Please explain more clearly what happen to the other eleven Pol II subunits.

Answer: To this end, we detected the levels and distributions of other 11 Pol II subunits. The immunofluorescent staining images indicated that distributions of other 11 Pol II subunits were not influenced when TANGO6 was suppressed. However, the western blot

results indicated that the protein levels of RPB3/6/11 reduced notably after knocking down TANGO6, while other subunits did not present obvious alterations. The results were presented and discussed in the revised manuscript (Extended Data Fig. 6; Line 297-302 and 472-481, marked by gray background). Thank you again for your constructive comments and suggestions.

Reviewer #2 (Remarks to the Author):

This manuscript by Feng et al. presents a study of the function of Tango6, a protein that has not been studied in detail before. The authors report that Tango6 is a transmembrane protein that localizes in the Golgi, is transported from the Golgi to the ER in COPI vesicles through interaction with COPI, interacts with NSF to undergo SNARE-mediated fusion with the nuclear ER, then strikingly jumps to solubility to enter the nucleus through the nuclear pore complex and becomes a nuclear protein. All the way, the authors claim, Tango6 is bound to RNA polymerase II subunit B (RPB2), and required to transport RPB2 into the nucleus. The transport of RNA polymerase II B by Tango6 would regulate the cell cycle, coupling a novel S-phase Golgi disassembly (that no one had seen before to my knowledge) to the entrance of RNA Polymerase II B, also a novel mechanism for regulation of cell cycle and G1/S transition.

There are many extraordinary claims here and one can get lost navigating the extended data to figure out what is possible, dubious, unsupported or outright impossible. The fact is that Tango6 localization is highly inconsistent among figures, the topology proposed for the protein is very dubious, basic cell biological mechanisms proposed (spectacularly novel in some cases) are presented with a confidence that is not commensurate with the strength of the evidence presented. Due to these and other many issues, I cannot recommend publication.

Thanks very much to your insightful estimation and valuable comments. Each issue was carefully considered and addressed by extra experiments. A panel of data were repeated and the descriptions and conclusions were intensively revised.

Main concerns:

1. How authors decide to study Tango6 needs more explanation. Line 87: "We screened a panel of molecules and identified Tango6... that displayed an identical cytoplasmic dynamic localization as COPA across the cell cycle" does not explain. The screening aim, candidates, approach and controls should be clearly presented.

Answer: Thanks for your valuable advice. Our lab is interested in the homeostasis of HSCs and hematopoiesis. To find key factors in the expansion of HSCs, we set up a simple screening system on cell lines by using a siRNA library (genOFF™ Human Genome siRNA Premix Library, Ribobio) to find key factors of membrane trafficking molecules in cell proliferation. At this stage, we screened 338 genes. Among them 223 genes influenced cell proliferation. TANGO6 attracted our attentions, because: 1) quite limited study and information was delivered in detail on this molecule. 2) the primary

immunofluorescent staining images indicated the interesting co-localization of TANGO6 with COPA in COPI vesicles and Golgi. We therefore decided to intensively investigate TANGO6. We are sorry for the unclear introduction and explanation on initiating TANGO6 study in our previous manuscript. The screening data and descriptions were included in the revised manuscript accordingly (Extended Data Fig. 2a-d; Line 97-107, highlighted by underline).

2. The authors state that Tango6 belongs to the TANGO family, but no such family exists. Tango designates a phenotypic class of genes showing defects in secretion upon knock down in Drosophila S2 cells (Bard et al.). Tango proteins are not evolutionarily or structurally related to each other. The direct nature of their effect on secretion is in many cases quite dubious: Tango 4 is spliceosome Prp19, Tango7 is translation factor eIF3m, Tango9 is a plasma membrane sugar transporter, Tango12 is a sodium pump. Is there a defect in secretion associated with Tango6 loss here?

Answer: Thank you very much for your suggestive questions. We learned more from your comments and we are quite sorry for the mistakes on the statement of TANGO family. We corrected the statement about TANGO6 accordingly (Line 67-71, highlighted by underline). To investigate the effects of TANGO6 on cell secretion, we knocked down TANGO6 to examine clathrin heavy chain (CLTC, a marker of clathrin), according to the reports that clathrin vesicle plays an essential role in cargo secretion^{1,2}. AP1M1 was used as a positive control³. The immunofluorescent staining images indicated that quite limited alteration of CLTC was detected upon TANGO6 deficiency by siRNA knock down. However, knocking down AP1M1 resulted in the diffuse of CLTC into the whole cells. Concordantly, the results from GSVA analysis disclosed that no notable differences regarding the secretory pathway were presented between *TANGO6^{WT}* and *TANGO6^{Mut}* cells. These data collectively indicated that TANGO6 played a limited role in cell secretion. The information was included in the revised manuscript (Extended Data Fig. 3k and 7h; Line 187-192 and 493-503, highlighted by underline).

3. The authors report a dispersion of Golgi COPI and cis-Golgi GM130 in S-phase that (to my knowledge) no one observed before. This should be carefully characterized, quantified and supported by cell cycle markers, as synchronization does not guarantee the specific cell shown is in a given phase of the cell cycle. In fact, DNA content from DAPI signal in images provided does not agree with S and G2 phase designations.

Answer: Thanks very much for your constructive comments. To further evaluate this assay, we collected the protein samples of G0, G1/S, S and G2 phases by serum starvation and double thymine block assay. Additional cell cycle markers, including TK1 (increase at late G1, and reaching the peak at S and G2 phases), and cyclin A (specifically expression at G2 phase), were applied. The western blot results indicated that TK1 levels gradually increased from G0 to G2 phase, while cyclin A was substantially existed at G2 phase (Extended Data Fig. 1a). The coordinated patterns of TK1 and cyclin A during cell cycle progression supported the synchronization assay. We re-examined the data of TANGO6 and COPA in previous Figure 1a. The result indicated the recapitulated patterns of TANGO6 and COPA. We realized that the previous DAPI signals in Figures were caused

by the collection time-points of later G2 phase (near M phase). We are sorry for this mistake and corrected the figures in the revised Fig. 1a. We additionally examined the patterns of GS28 (*cis*-Golgi marker), Giantin (*mid*-Golgi marker) and Golgin97 (*trans*-Golgi marker). The immunofluorescent staining data indicated that GS28⁺ and Giantin⁺ signals were aggregated at G0 and G1/S phases, diffused to the cytoplasm at S phase and condensed again at G2 phase. However, Golgin97⁺ signals limitedly changed during cell cycle progression. Combined with the repeated GM130 (*cis*-Golgi marker) and TGN46 (*trans*-Golgi marker) data, we concluded that *mid/cis*-Golgi disassembled dynamically, while *trans*-Golgi was limitedly altered during cell cycles. The new data and information were included accordingly (Extended Data Fig. 1b-h; Line 87-97 and 508-517, highlighted by underline).

4. Is this a whole Golgi disassembly or specific to COPI/GM130? The authors seem to suggest with *trans*-Golgi network marker TGN46 that this is only COPI and GM130, but can they show in the same image that COPI or GM130 are dispersed while the structure of the Golgi remains unperturbed (TGN as well, but better with more integral *cis*, *mid* and *trans* markers).

Answer: Thanks very much for your valuable suggestion. Accordingly, we examined ERGIC53 (ER-Golgi intermediate compartment marker), Giantin (*mid*-Golgi marker), GS28 (*cis*-Golgi marker) and Golgin97 (*trans*-Golgi marker) in the cell cycle progression assays by the immunofluorescent staining. The results indicated that the patterns of Golgin97⁺ signals (*trans*-Golgi marker) were limitedly altered. However, ERGIC53⁺, Giantin⁺ and GS28⁺ signals were aggregated at G0 and G1/S phases, which then diffused to the cytoplasm at S phase and condensed again at the G2 phase. These results indicated the disassembly of *cis*-Golgi, *mid*-Golgi but not *trans*-Golgi during cell cycles. Considering the Golgi and ER trafficking pathway, we believe that *mid/cis*-Golgi and COPI vesicle disassembled dynamically, while *trans*-Golgi, ER and COPII vesicle are limitedly altered during the interphase of cell cycles. The interesting phenotypes of Golgi and ER in cell cycle warrant further investigate. We included these data in the Extended Data Fig. 1b-i. The descriptions and discussion were included in line 89-99 and 508-517, highlighted by underline.

5. The authors claim that Phobius and UNiprot predict six transmembrane domains and one signal peptide in Tango6. However, I only saw one possible transmembrane domain and no signal peptide in these predictions. Could the authors clarify this?

Answer: Thanks for your careful evaluation. We initially used Phobius and UNiprot to predict the transmembrane domains. As you see, these webs suggested one possible transmembrane domain. However, we were curious on how Tango6 carried RPB2, if only one transmembrane domain existed. Therefore, we additionally tried several possible websites, including Phyre2 and PSIPRED. Although Phobius and Uniprot suggested one putative transmembrane domain at 472-487 aa, the Phyre2 and PSIPRED websites suggested an additional putative signal peptide at 1-39 aa and other five possible transmembrane domains (see the followed figure 1). We verified all these predictions and eventually identified the real transmembrane domains (146-161 aa and 472-487 aa) and a

signal peptide. We are quite sorry for the incomplete information in the previous manuscript. We included the information in our revised manuscript (Line 136-140, highlighted by underline)

Figure 1. The putative transmembrane domains and signal peptide of TANGO6 by Phyre2 (left) and PSIPRED (right).

6. The topology of the protein is in fact extremely unclear. If Tango6 has a signal peptide, how come all three constructs in Fig 1f show the same localization, including nuclear puncta in all three? Signal peptide-DsRed should be just secretable DsRed (after signal peptide cleavage). And how come both N and C termini end up cytoplasmic? Shouldn't HA be cleaved from HA-Tango6-FLAG?

Answer: Thanks very much for your careful evaluation and comments. The nuclear puncta in previous Fig 1f was probably caused by a high dose of plasmid delivery and a strong exposure of imaging. We are quite sorry for this mistake. We repeated the experiments by transfecting low volume of plasmids. The image results indicated that "Signal peptide-DsRed, TM2-DsRed and TM5-DsRed" presented the same localization in the cytoplasm but no notable nuclear puncta were presented. The revised data was included in the new Fig.2c. To address the secretory possibilities of the signal peptide (SP), we designed constructs with HA tag behind or before signal peptide (Extended Data Fig 3a). IGF1 was selected as a positive control, because the signal peptide of IGF1 could be cleaved and secreted⁴. The constructs of SP-HA-TANGO6(Δ SP) / HA-SP-TANGO6(Δ SP) and SP-HA-IGF(Δ SP)/HA-SP-IGF1(Δ SP) were designed and delivered. The western blot results indicated that HA signals were not observed in the cell cytoplasm after transfecting SP-HA-IGF(Δ SP)/HA-SP-IGF1(Δ SP) plasmids, in contrast to clear appearance after transfecting SP-HA-TANGO6(Δ SP)/HA-SP-TANGO6(Δ SP) simultaneously (Extended Data Fig 3b). This result indicated the non-secretory natures of signal peptide in TANGO6. To further confirm the conclusion, we designed the SP-HA-TANGO6(Δ SP)-FLAG plasmids (Extended Data Fig 3c) and delivered them to the cells. HA and FLAG signals could be comparably detected in the groups after transfecting, which recapitulated the patterns of TANGO6 after treated by Digitonin and Triton-X-100 (Extended Data Fig 3g). These results indicated that the signal peptide of TANGO6 was not cleaved. We reviewed previous studies and found that some factors, such as MP19, was not secretory and mainly localized at cell membrane⁵. The new data and information were included and discussed in the in the revised manuscript (Extended Data Fig. 3a-c, g; Line 147-165 and 493-503, highlighted by underline).

7. The study relies heavily in antibody stainings and subcellular localization studies. However, the localization of Tango6 is highly inconsistent among images. Tango6 antibody, HA-Tango6, HA-Tango6-Flag and Tango6-mMaple3 displayed completely different localization pattern. For example: compare anti-Tango6 in Fig 1a versus Fig S2f. Also HA-Tango6 in Fig2k, Tango6-mMaple3 in Fig S4b. One has to wonder whether Tango6 signals in the nucleus are real or typical antibody staining unspecific specks, or belong to a plane that's not inside the nucleus. Simultaneous anti-Tango6 (green) /anti-FLAG or HA (red) in Fig. 1g does not show good colocalization when checked in detail. Importantly, when stained together in Fig 2n, Tango6 and RPB2 show very little colocalization.

Answer: Thanks very much for your careful evaluation and suggestions. We are sorry for the different TANGO6 distribution in previous Fig. 1a and S2f. We realized that the images in previous Fig S2f were overexposed. We repeated this data with appropriate imaging in the revised Fig. 1g. For other pictures, the inconsistent localization between different images was probably due to the antibody staining methods. The TANGO6 antibodies targeted the endogenous TANGO6 proteins in the cells. However, the HA-Tango6, HA-Tango6-Flag and Tango6-mMaple3 staining was performed after delivery of extra plasmids. We found that the exogenous TANGO6⁺ signals were distributed in the whole cell cytoplasm and nucleus after transfecting high volumes of plasmids. However, when low dosage of plasmids was delivered, the consistent co-localization between HA-TANGO6-FLAG and TANGO6 (endogenous) were discerned. To clearly define the vital site of NLS in the nuclear entry of cytoplasmic TANGO6⁺ signals, we transfected high dosages of plasmids (300 ng/well), which led to the inconsistent patterns of HA-Tango6 and Tango6-mMaple3 with that of TANGO6 in previous Fig. 2k. To further elucidate this issue, we repeated this assay by providing low dosage of plasmids (100 ng/well). The results indicated the consistent signal patterns of HA-TANGO6-FLAG and TANGO6. The data was included in the revised Fig. 3k.

To address “whether Tango6 signals in the nucleus are real”, we separated cytoplasm and nucleus to conduct western blot. The results indicated the clear TANGO6 bands in the samples from the nucleus. The TANGO6 sole signals in Fig 1g were probably due to the transfect efficiency of exogenous TANGO6. However, the condensed signals of endogenous TANGO6⁺ were co-localized with exogenous HA-TANGO6-FLAG.

Indeed, there existed non-colocalized signals of Tango6 and RPB2 in previous Fig 3n. Because the MS data disclosed a panel of factors interacting with Tango6, and there should exist additional factors to facilitate the nuclear entry of RPB2. We speculate that the TANGO6 or RPB2 only signals are probably these additional factors. However, we repeated the experiments and the data indicated the large co-localization of TANG6 with RPB2. The new data was provided in the revised Fig. 4n. Our current study focused on the roles of TANGO6 in binding RPB2. The Co-IP and double staining data manifested their interaction and co-localization. We are sorry for lacking the information regarding the only signals in previous Fig 2n. We included the discussion in the revised manuscript (Line 523-525, highlighted by underline).

8. If Tango6 has 6 hydrophobic transmembrane regions, how does it come off membrane, passes the nuclear pore and just exists floating inside the nucleus? This cannot be simply

stated without clear proof.

Answer: Thanks very much for this interesting question. This issue also puzzled us for a long time. Our data indicated that TANGO6 harbored two transmembrane domains. We previously proposed that the fragments of 161-472aa were in the COPI vesicle lumen and its N- and C- terminus in cell cytoplasm. However, we recently double-checked our data in detail and realized another feasible topology that the segment (161-472 aa) was located in the cytoplasm. To weight these two models, we constructed a TM1-HA-F(fragment 161-472 aa)-FLAG-TM2 plasmid by fusing HA tag behind TM1 motif and FLAG tag before TM2 motif. The immunofluorescent staining images indicated that both HA⁺ and FLAG⁺ signals were detected in either the digitonin or Triton X-100 treated groups (Fig. 2g). This result suggested that the segment of 161-472 aa locating in the cell cytoplasm (Extended Data Fig. 3j). We think that this topology of TANGO6 are reasonable, because the TMs might superficially immersed in the COPI membrane, which is easy to be cut off from the membrane by SNARE protein mediated membrane fusion manner⁶. We are quite sorry for our initial mistakes on the topology of TANGO6. The corrected models and descriptions were included and discussed accordingly (Fig. 2h; Line 176-187 and 486-492, highlighted by underline).

To explore the mechanism of nuclear entry of TANGO6 from ER, LRRC59, an ER resident protein that mediated membrane protein nuclear import⁷, was paid attention to. CO-IP result indicated an obvious interaction between TANGO6 and LRRC59. The immunofluorescent staining results indicated a significant accumulation of TANGO6⁺ signals at cell cytoplasm but reduced intensities in the nucleus after knocking down LRRC59. However, TANGO6 protein levels and ER integrity (marker by calnexin) were not affected under this condition (Fig. 3f and Extended Data Fig. 4g-j; Line 224-235, highlighted by underline).

9. This study wants to demonstrate that Tango6 affects cell cycle by mediating RPB2 nuclear transport. Missing experiments: RPB2 deficiency phenotype on cell proliferation; RPB2 supplement rescues Tango6 mutant.

Answer: We appreciate your suggestions. To this end, we constructed a plasmid by fusing a NLS (Pro-Lys-Lys-Lys-Arg-Lys-Val) at N- terminus of HA-RPB2 to perform the rescue experiments. The immunofluorescent staining result indicated that only NLS-tagged HA-RPB2 was detected in *TANGO6*^{KO} cells, while HA-RPB2 was not. Ki-67 staining was conducted. The results showed that supplement of NLS-tagged HA-RPB2 rescued the proliferation deficiency phenotypes in *TANGO6*^{KO} cells (Fig. 5f and Extended Data Fig. 7k; Line 369-377, highlighted by underline).

Other points:

1. The ratio of COPI colocalizing with scattered Tango6 is obviously much lower than 60% in Fig1c.

Answer: We are sorry for the mistakes in the ratio statistics. We previously counted the ratios of COPA⁺ signals co-localized with TANGO6⁺ signals in the total COPA⁺ signals in the images. We realized that this counting method was wrong. We re-counted the

percentage of TANGO6⁺ signals merged with COPA⁺ signals among the TANGO6⁺ signals. And the ratio is approximately 23%. We clarified this issue in the revised manuscript accordingly (Fig. 1d; Line115-116, highlighted by underline).

2. Line215. The authors conducted FLAG-Tango6 pull down-MS experiments and identified RPB2. First, controls are missing here. Second: in previous results, Tango6 showed strong interactions with COPI and NSF, which are missing in MS results.

Answer: We used IgG-beads as an endogenous control and empty pCS2 plasmid as the exogenous control to perform the MS experiments. COPB1 (a COPI vesicle component) and NSF were indicated in the data of exogenous FLAG-Tango6 group but not MS group. We suspected that the levels of COPB1 and NSF in the MS group were not enough to be detected, in compared to that from the FLAG-Tango6 group. We highlighted COPB1 and NSF in the MS results by blue colors in the revised datasheet.

3. The authors claim nuclear import of Tango6 protein in Thapsigargin treatment (2f) and KPNB1 KD (2n) goes down, but overall signal levels are also very reduced. Furthermore, there wasn't any nuclear Tango6 signal in Fig. 1a (see major point 8).

Answer: Thanks for your careful estimation. To further evaluate the alterations of Tango6 protein levels in the cytoplasm and nucleus after Thapsigargin treatment and KPNB1 KD, we isolated TANGO6 proteins from the cytoplasm and nucleus respectively to perform western blot. The result indicated that the cytoplasmic TANGO6 protein levels were limitedly reduced but the nuclear counterparts decreased significantly after Thapsigargin treatment and KPNB1 KD. We included the data in the revised manuscript (Extended Data Fig. 4f and m).

4. The authors omit from their analysis the M phase of the cell cycle, which is when the nuclear membrane breaks down and the Golgi really disassembles. What happens with RPB2 then?

Answer: Thanks for this interesting question. We examined the patterns of RPB2 in the M phase accordingly. The immunofluorescent staining results (see Figure 2 below) indicated that the majority of RPB2 localized at the cell nucleus at prophase. However, RPB2 largely concentrated at cell cytoplasm at metaphase and anaphase, and re-appeared at nucleus and cytoplasm at telophase.

Figure 2. The distributions of RPB2 in prophase, metaphase, anaphase and telophase. Scale bar, 10 μ m

5. Line293. In the heatmap of Tango6 KO vs WT (Fig S6C), the trend of RPB2/3/4/8/9/10 reduction is not clear, and WT-1 is so different compared to others. Furthermore, this result is inconsistent with results shown in Fig S5k and Fig S5n, where RPB2 level is unchanged with Tango6KD.

Answer: We deeply apologized for the mistakes that the gene names of RPB subunit members were presented in a wrong order in the previous heatmaps. We corrected it in the revised manuscript (Extended Data Fig 7c). To validate this result, qRT-PCR was conducted to detect the transcript levels of 12 subunits. The results indicated that the levels of RPB2/3/4/10 reduced obviously (Extended Data Fig 7d). We repeated the west blot data in previous Fig S5k. The results indicated a notable decrease of RPB2 levels upon TANGO6 KD (Extended Data Fig 5m). Previous Fig S5n presented RPB2 distribution after transfecting HA-RPB2 plasmid. And this assay did not reflect the pure endogenous RPB2 protein levels but also the extra-delivered ones. The data and conclusions were included in the revised manuscript (Extended Data Fig. 5n, 7c-d; Line 359-364; highlighted by underline).

6. In FigS5k and FigS5m, Tango6 KD show different effect on RPB2 in U251 cells.

Answer: Previous FigS5k indicated the TANGO6 levels after transient KD by siRNA, whereas FigS5m presented TANGO6 levels in the mutant cell lines that deleted TANGO6 genomic sequences. To further confirm these results, we repeated the KD assay to perform the western blot. The result showed that RPB2 was significantly reduced but still detectable in the KD-TANGO6 (Extended Data Fig. 5m). However, the RPB2 protein levels were notably reduced in the KD group but not so dramatic when compare with KO groups (Extended Data Fig. 5k). The different effects of RPB2 in both assays should be due to the efficiency of TANGO6 depletion by transient KD and permanent KO.

7. In Tango6 cKO mice, RPB2 protein level decreases. However, if Tango6 mediates RPB2 import into nucleus, without Tango6 the RPB2 amount should be unchanged, while showing aggregation outside nucleus, like shown in HeLa cell (Fig 3c, 3d, 3i). Is the expression/protein level of RPB2 affected or not by Tango6 deficiency?

Answer: That is an interesting query. To address this issue, we sorted Lin⁻Sca-1⁺Kit⁺ (LSK) cells from E14.5 mice fetal liver to perform qRT-PCR. The results indicated that *Rpb2* mRNA level was reduced significantly (Extended Data Fig. 8r), indicating that Tango6 deficiency suppressed RPB2 expression. We speculate that RPB2 is a key component involved in RNA polymerase II mediated mRNA transcription. The impaired nuclear import of RPB2 in the cKO mice caused a series of compromised genes transcription during embryos development, which might influence RPB2 itself. This question was quite interesting for the future study. We included this information in the revised manuscript and thanks very much again (Extended Data Fig. 8r; Line 439-441, highlighted by underline).

8. Line360. The authors conclude that Tango6, RPB2 and COPI have synergistic effect on G1/S progression. More experiments (such as double KD) are required to say they have synergistic effect on G1/S progression.

Answer: Thanks for your suggestion. We performed the double knock-down experiments accordingly. Western blot of TK1 (S phase marker) and Cyclin A (G2 phase marker) and immunofluorescent staining of Ki-67 indicated that reduced cell proliferation was observed when any two factors (KD-TANGO6/RPB2, KD-TANGO6/COPA and KD-RPB2/COPA) were knocked down. Double KD of RPB2 and COPA (KD-RPB2/COPA) made the most conspicuous effect on G1/S progression among them. The data was included (Fig. 5l, m and Extended Data Fig. 7t; Line 386-391; highlighted by underline).

9. The target of Tango6 identified here is RNA polymerase II. So why cells with Tango6 deficiency arrest at G1/S phase, instead of any other phase?

Answer: Thanks very much for the interesting question. CO-IP experiment revealed that the interaction between TANGO6 and RPB2 was the strongest at the G1/S phase in the interphase of cell cycles. The western blot results indicated that the levels of RPB2 increased from G1/S phase and obviously enhanced in S and G2 phases, especially in the nucleus (Fig. 3a, 4m, and Extended Data Fig. 1a). These results implied that TANGO6 captured RPB2 largely at G1/S phase. Therefore, a possible explanation for this phenotype is that TANGO6 deficiency compromises RPB2 nuclear entry and leads to cell cycle largely arrest at G1/S phase. We included this information in the revised manuscript (Discussion, line 535-541, highlighted by underline).

10. Figures 4 and 5, presenting loss of function of Tango6, show cell proliferation defects in conditional KO mice, but this is not a terribly specific phenotype, it is compatible with animals and cells just being sort of sick, and therefore the results do not add much support to the proposed Tango6/COPI/SNARE/NPC/RNA Pol II/cell cycle axis.

Answer: Thanks very much to your comments. The data from animal models of cKO mice was used to support the application of TANGO6/COPI/PolII in the *in vivo* regulation of HSCs proliferation. However, to provide more evidence of Tango6 in regulating the hematopoietic phenotypes *in vivo*, we examined the artery that gave birth to HSCs at the embryonic stages. The imaging results indicated a comparable artery between cKO mice and WT controls (Extended Data Fig. 8 i and j), indicating the emergency of HSCs was not affected in cKO mice but their expansion was compromised. To further valuate these phenotypes, the HSCs (LSK cells) were sorted from the bone marrow of cKO mice to be cultured. BFA and packaged RPB2 lentivirus were delivered to the cultured HSCs. The followed observation manifested that HSCs expansion was significantly affected, suggesting disrupting COPI vesicle trafficking and RPB2 activity led to a similar defect proliferation of HSCs to that in the cell lines. These phenotypes supported the employ of Tango6/COPI/RNA Pol II in HSCs expansion. The data and conclusions were included in the revised manuscript (Extended Data Fig. 8 t; Line 409-421, marked by light green background).

Reviewer #3 (Remarks to the Author):

The work has high significance with potential identification of intra-organelle transport

system for critical cell cycle machinery such as RPB2. The conclusions are supported by the in vitro results.

However, the analysis focusing on the fetal liver expansion of HSCs and the role that TANGO6 plays in this specific aspect of hematopoietic development needs to be further and more carefully investigated. Specifically the flow cytometry analysis of the fetal liver must be conducted using LSK and SLAM markers (CD150+CD48-) as the use of only LSK progenitors markers can be confounded by primitive waves (yolk sac) of developmental hematopoiesis which are not affected by the vavCre driver. Furthermore, while it is commendable that the authors attempted a tissue specific deletion of TANGO6 after the constitutive mutant was embryonic lethal at a very early stage, the concern is that that even the vaviCre mutant appears to have a dramatic phenotype that precludes genuine hematopoietic analysis in the fetal liver. The defect observed in Fig.5c-e is striking and indicates a complete loss of hematopoietic cellularity, so since VavCre is predicted to be active in the hematopoietic compartment starting at ~E12 (Joseph C et al Cell Stem Cell 2013) but can also affect endothelial cells. This would explain the severe pale embryos of the TANGO6 conditional knockouts. Thus the authors must also analyze the emerging HCSs in the aorta gonad mesonephos (AGM) of their mutant mice in order to determine if loss of TANGO6 affects the formation of the HSCs or just their proliferation. Assessment of the vascular tissue at E9.5, E12.5 and E14.5 by staining for VeCadherin and PECAM1 is also required to rule out any defect in vascular development which may preclude any analysis of the fetal liver hematopoiesis. Since TANGO6 is predicted to affect proliferation, angiogenic developing vasculature may also be drastically affected and in fact the pale vaviCre+ embryo in Fig.5c suggests an defect during earlier development than the fetal liver. If this is the case, apoptosis markers should also be assessed along with the proliferation analysis, since a defective vascular tissue will negatively impact hematopoietic viability.

Thanks very much for your careful comments and valuable suggestions. We performed extra experiments to address your concerns in the followed responses.

1. Specifically the flow cytometry analysis of the fetal liver must be conducted using LSK and SLAM markers (CD150+CD48-) as the use of only LSK progenitors markers can be confounded by primitive waves (yolk sac) of developmental hematopoiesis which are not affected by the vavCre driver.

Answer: Thank you very much for your kind advice. Accordingly, we used SLAM markers to perform the flow cytometry analysis. The results indicated that both LT/ST-HSC fractions were reduced at E15.5, indicating that TANGO6 is indispensable for embryonic HSCs proliferation in mice. The data and information was included (Extended Data Fig. 8m,n; Line 426-428; marked by light green background).

2. Thus the authors must also analyze the emerging HCSs in the aorta gonad mesonephos (AGM) of their mutant mice in order to determine if loss of TANGO6 affects the formation of the HSCs or just their proliferation.

Answer: To address this issue, we examined Runx1+ cells in AGM regions at E9.5 when

HSCs emerged there⁸. The immunofluorescent staining images indicated comparable Runx1⁺ signals between WT and cKO mice at this stage, suggesting HSCs emergence was not affected upon loss of TANGO6 in the cKO mice (Extended Data Fig. 8k; Line 419-421; marked by light green background). Thanks very much!

3. Assessment of the vascular tissue at E9.5, E12.5 and E14.5 by staining for VeCadherin and PECAM1 is also required to rule out any defect in vascular development which may preclude any analysis of the fetal liver hematopoiesis.

Answer: We appreciate your wonderful advice. We assessed the vascular tissue at E9.5, E12.5 and E14.5 by staining PECAM1 and VE-cadherin as suggested. The immunofluorescent staining images indicated that PECAM1⁺ signals did not present notable alterations in the cKO mice in compare to the WT controls during E9.5-E12.5. However, VE-cadherin⁺ signals were comparable between the cKO and WT controls at E9.5 but reduced obviously in E12.5 and E14.5 cKO mice than the WT controls (Extended Data Fig. 8i,j). These data indicated that blood vessel was not notably affected at the early stages when HSCs were produced. The compromised expression of VE-cadherin but not PECAM in the cKO mice was an interesting phenotype warranting further investigation. The data and discussions were included accordingly (Line 409-418 and 550-558, marked by light green background).

4. If this is the case, apoptosis markers should also be assessed along with the proliferation analysis, since a defective vascular tissue will negatively impact hematopoietic viability.

Answer: We examined Active-Caspase 3 at E14.5. The results indicated that no obvious difference of Caspase 3⁺ signals was discerned between WT and cKO mice (Extended Data Fig. 8q; Line 432-434; marked by light green background). Thanks very much for all your concerns.

Reference

- 1 Pley, U. & Parham, P. Clathrin: its role in receptor-mediated vesicular transport and specialized functions in neurons. *Crit Rev Biochem Mol Biol* 28, 431-464, doi:10.3109/10409239309078441 (1993).
- 2 Neumann, U., Brandizzi, F. & Hawes, C. Protein transport in plant cells: in and out of the Golgi. *Ann Bot* 92, 167-180, doi:10.1093/aob/mcg134 (2003).
- 3 Kinghorn, K. et al. A defined clathrin-mediated trafficking pathway regulates sFLT1/VEGFR1 secretion from endothelial cells. *bioRxiv*, doi:10.1101/2023.01.27.525517 (2023).
- 4 Bai, W. L. et al. IGF1 mRNA splicing variants in Liaoning cashmere goat: identification, characterization, and transcriptional patterns in skin and visceral organs. *Anim Biotechnol* 24, 81-93, doi:10.1080/10495398.2012.750245 (2013).
- 5 Chen, T., Li, X., Yang, Y. & Church, R. L. Localization of lens intrinsic membrane protein MP19 and mutant protein MP19(To3) using fluorescent expression vectors. *Mol Vis* 8, 372-388 (2002).
- 6 Sudhof, T. C. & Rothman, J. E. Membrane fusion: grappling with SNARE and SM

- proteins. *Science* 323, 474-477, doi:10.1126/science.1161748 (2009).
- 7 Zhen, Y. et al. Nuclear import of exogenous FGF1 requires the ER-protein LRRC59 and the importins Kpnalpha1 and Kpnbeta1. *Traffic* 13, 650-664, doi:10.1111/j.1600-0854.2012.01341.x (2012).
- 8 Gao, X., Xu, C., Asada, N. & Frenette, P. S. The hematopoietic stem cell niche: from embryo to adult. *Development* 145, doi:10.1242/dev.139691 (2018).

REVIEWER COMMENTS

Reviewer #2 (Remarks to the Author):

The authors have conducted many experiments and observations to address my concerns, resulting in a revised topology for the protein and many other changes to the manuscript. If the signal peptide is not cleaved it should perhaps be called a signal anchor.

Remaining concerns and comments:

A. Regarding the dispersion of Tango6 and COPI in S-phase, I asked in comment 4 if it would be possible to image simultaneously Golgi+Tango6 and Golgi+COPI. This would lend support to the existence of this proposed cis-Golgi disassembly that no one had seen before. I also asked for some quantification of the cis-Golgi disassembly, rather than taking single images of cell cycle-synchronized cells as sufficient evidence of this remarkably novel phenomenon.

B. The question in comment 2 as to whether Tango6 deficiency affects secretion (to confirm in mammals the initial Tango phenotype in Drosophila S2 cells) remains unanswered. The authors conclude that it doesn't based on very weak arguments. This would be surprising and in fact cancel the rationale for including Tango6 in a library of secretory genes. The first argument about clathrin being unchanged (Line 187-192) makes little sense: clathrin and AP1 may affect Golgi-to-PM traffic, but not ER-to-Golgi. The second argument, about expression of secretory pathway genes being unchanged, is very lateral (Line 498: "RNA-seq data indicated the limited alteration of the secretory pathway between the mutant and WT cells"). Instead, direct observation of any retained cargo inside of cells and/or its absence in supernatants would answer this.

C. This statement in the added text should be revised:

Line 67: "The transport and Golgi organization (TANGO) designates a phenotypic class of factors located in the Golgi apparatus in Drosophila S2 cells; the disruption of the transport of these factors affects secretion".

I copy here from my comment 2: “The authors state that Tango6 belongs to the TANGO family, but no such family exists. Tango designates a phenotypic class of genes showing defects in secretion upon knock down in Drosophila S2 cells (Bard et al.)”.

D. The authors added this text:

Line 69: “Although the roles of TANGO1/2/4/7/9/12 have been investigated[17-22]...”

In fact, only Tango1 and 2 have been investigated in some detail. My comment about Tango4 being a spliceosome protein, Tango7 a translation factor, 9 a sugar transporter and 12 a sodium pump was not meant to be exhaustive, but just to exemplify how Tango proteins were not a molecular family.

E. According to the description now supplied by the authors (lines 100-107), the study should not be presented as a screening but as a targeted study of Tango6.

Reviewer #3 (Remarks to the Author):

The authors have been responsive and have attempted to perform all the suggested experiments. The remaining concerns with respect to the hematopoietic and vascular analysis are listed below:

1). The authors have not improved the quality of their findings in the main figures, and their new more careful analysis of the FL HSCs (SLAM markers) is in the Extended Data Fig.8 rather than in the main figures. If the Tango6 K/O is affecting the FL HSCs then this data should be of higher quality and included in the main figures. The LSK progenitor result is not as impactful and not very convincing. High quality data of the FL HSCs should be included in the main manuscript.

2). The quality of the data shown in Extended Fig.8 with regard to both the HSC and vascular analysis is poor. Specifically the authors show that the HSCs are decreased in ExFig.8m but the actual representative FACS plot is not shown for the Tango6 k/o mutant. Then for the

Runx1 staining in the AGM, the authors do not also stain for PECAM1 thus it is impossible to determine which cells are actually expressing Runx1 because there is no identifying marker of the tissue. Finally how is it possible that an embryo that appears to be completely pale and to have to have no vascular structures as shown in Fig.6c and ExFig.8h has "equal numbers of PECAM1+ cells at E12.5". This is not explained by the authors and no new data tries to address this issue.

3. Lower magnification images are required for the AGM and FL in order to allow the reader to orient themselves to these tissues. It is impossible to determine where any of the sections shown in ExFig.8i, j, k are located and their tissue of origin. The AGM and FL are very different in structure and orientation so this can be improved by a lower magnification image.

REVIEWER COMMENTS

Reviewer #2 (Remarks to the Author):

The authors have conducted many experiments and observations to address my concerns, resulting in a revised topology for the protein and many other changes to the manuscript. If the signal peptide is not cleaved it should perhaps be called a signal anchor.

Thanks very much for your valuable advices. We corrected the description of signal peptide to signal anchor. We carefully addressed the remaining concerns as suggested. The information was included in the revised manuscript accordingly.

Remaining concerns and comments:

A. Regarding the dispersion of Tango6 and COPI in S-phase, I asked in comment 4 if it would be possible to image simultaneously Golgi+Tango6 and Golgi+COPI. This would lend support to the existence of this proposed cis-Golgi disassembly that no one had seen before. I also asked for some quantification of the cis-Golgi disassembly, rather than taking single images of cell cycle-synchronized cells as sufficient evidence of this remarkably novel phenomenon.

Answer: We deeply appreciate your constructive comments and suggestions. To this end, we simultaneously imaged the patterns of Golgi (GM130) and TANGO6/COPI (COPA) at one (early stages), four (middle stages) and six (late stages) hours post eliminating thymidine in the S phase. The immunofluorescent staining images indicated that the GM130⁺ signals were condensed initially (01:00). Then, these signals dispersed into spots (04:00) and stacks (06:00). The patterns of TANGO6⁺ and COPA⁺ signals were identical with those of GM130 at both time points. The information was included the revised manuscript (Extended Data Figs. 1m and n, Line 115-116, marked by gray background).

We imaged the dynamic distributions of GM130⁺ signals (*cis*-Golgi) during S phase (00:00-07:00) via characterizing HeLa cells at sequential time points accordingly. Clearly, the GM130⁺ signals present different behaviors: 1) The GM130⁺ signals are condensed at 00:00, which expand at 01:30. 2) The appearance of GM130⁺ signals changes into the isolated stacks at 03:00; 3) The isolated stacks disassemble into Golgi hazes like structures at 04:30; 4) The Golgi hazes like structures increase and re-converge at 07:00. We quantified the total and average areas of GM130⁺ signals (μm^2) according to a previous report¹. The statistical graph indicated that the total areas of GM130⁺ signals were increased from 111.60 μm^2 to 248.30 μm^2 at 00:00 and 04:30. They then reduced to 150.80 μm^2 at 07:00. However, the average area of GM130⁺ signals reduced from 2.22 μm^2 to 0.64 μm^2 at the same time window, which recovered to 0.96 μm^2 at 07:00. The data were included accordingly (Extended Data Figs. 1j-l, Line 95-101, marked by gray background).

B. The question in comment 2 as to whether Tango6 deficiency affects secretion (to confirm in mammals the initial Tango phenotype in Drosophila S2 cells)

remains unanswered. The authors conclude that it doesn't based on very weak arguments. This would be surprising and in fact cancel the rationale for including Tango6 in a library of secretory genes. The first argument about clathrin being unchanged (Line 187-192) makes little sense: clathrin and AP1 may affect Golgi-to-PM traffic, but not ER-to-Golgi. The second argument, about expression of secretory pathway genes being unchanged, is very lateral (Line 498: "RNA-seq data indicated the limited alteration of the secretory pathway between the mutant and WT cells"). Instead, direct observation of any retained cargo inside of cells and/or its absence in supernatants would answer this.

Answer: Thanks very much for your valuable instructions. Accordingly, we collected the proteins from HeLa cell lysate and medium supernatants after knocking down TANGO6 and AP1M1, a key regulator of clathrin-associated protein complex to regulate secretory², as the positive control. EGF and Collagen I were examined, because they are the typical secretory cargos^{3,4}. The western blot results indicated that quite limited alteration of EGF and Collagen I levels was observed in both the cell lysate and supernatant upon TANGO6 knockdown. However, interfering AP1M1 caused a significant aggregation of EGF and Collagen I inside the cells and obvious reduction in the supernatants. These results support the conclusion that TANGO6 played a limited role in protein secretory. The revised data were included accordingly (Extended Data Figs. 3I-n; Line 198-206 and 507-511, marked by gray color).

C. This statement in the added text should be revised:

Line 67: "The transport and Golgi organization (TANGO) designates a phenotypic class of factors located in the Golgi apparatus in Drosophila S2 cells; the disruption of the transport of these factors affects secretion".

I copy here from my comment 2: "The authors state that Tango6 belongs to the TANGO family, but no such family exists. Tango designates a phenotypic class of genes showing defects in secretion upon knock down in Drosophila S2 cells (Bard et al.)".

Answer: Thank you for your kind advice. We corrected the statement in the revised manuscript (Line 67-68, marked by gray color).

D. The authors added this text:

Line 69: "Although the roles of TANGO1/2/4/7/9/12 have been investigated[17-22]..."

In fact, only Tango1 and 2 have been investigated in some detail. My comment about Tango4 being a spliceosome protein, Tango7 a translation factor, 9 a sugar transporter and 12 a sodium pump was not meant to be exhaustive, but just to exemplify how Tango proteins were not a molecular family.

Answer: Thanks for your kind comments. We are quite sorry for the mistakes on the

statement of the investigation range of TANGO. The revised introduction was included accordingly (Line 69-70, marked by gray color).

E. According to the description now supplied by the authors (lines 100-107), the study should not be presented as a screening but as a targeted study of Tango6.

Answer: We corrected information in our revised manuscript (Line 103-104, marked by gray color). Thank you again for your constructive comments and suggestions.

Reviewer #3 (Remarks to the Author):

The authors have been responsive and have attempted to perform all the suggested experiments. The remaining concerns with respect to the hematopoietic and vascular analysis are listed below:

Thanks very much for your careful comments and valuable suggestions. We performed extra experiments to address your concerns.

1). The authors have not improved the quality of their findings in the main figures, and their new more careful analysis of the FL HSCs (SLAM markers) is in the Extended Data Fig.8 rather than in the main figures. If the Tango6 K/O is affecting the FL HSCs then this data should be of higher quality and included in the main figures. The LSK progenitor result is not as impactful and not very convincing. High quality data of the FL HSCs should be included in the main manuscript.

Answer: Thank you very much for your suggestive comments. We examined the LSK cells in the fetal liver at E13.5-E16.5 and SLAM pools (LT/ST-HSCs) at E15.5 by performing flow cytometry. The result indicated that LSK (Lin⁻Sca-1⁺c-Kit⁺) fractions were comparable between WT and cKO mice at E13.5, which reduced drastically from E15.5 in the cKO mice. However, both LT-HSC and ST-fractions in the isolated CD150⁺CD48⁻ pools of LSK populations were reduced at E15.5 in cKO mice. We tried our best to improve the data quality, which was included in the main figures as suggested. The revised data were included in Figures. 6f-i.

2). The quality of the data shown in Extended Fig.8 with regard to both the HSC and vascular analysis is poor. Specifically the authors show that the HSCs are decreased in ExFig.8m but the actual representative FACS plot is not shown for the Tango6 k/o mutant. Then for the Runx1 staining in the AGM, the authors do not also stain for PECAM1 thus it is impossible to determine which cells are actually expressing Runx1 because there is no identifying marker of the tissue. Finally how is it possible that an embryo that appears to be completely pale and to have to have no vascular structures as shown in Fig.6c and ExFig.8h has "equal numbers of PECAM1⁺ cells at E12.5". This is not explained by the authors and no new data tries to address this issue.

Answer: We appreciate your constructive comments. We are quite sorry for missing the actual representative FACS plot of cKO mice in previous Extended Fig. 8m. We added the

data in the revised main figures (Fig. 6h). To estimate the Runx1 signals and HSC emergence in the AGM, we initially performed the double immunofluorescent staining of RUNX1 (rabbit) with PECAM (mice) as suggested. However, we tried several primary anti-bodies of PECAM1 of mice reagents and the staining data was poor. To this end, we performed the double staining using VE-cadherin, because VE-cadherin and PECAM1 presented the identical labels of endothelial cells and these two markers did not show notable alteration between WT and cKO mice at E9.5 (Extended Data Figs. 8i and j). The results revealed the limited changes of VE-cadherin⁺ and Runx1⁺ signals in the AGM regions between WT and cKO mice, indicating that the HSCs emergence was not obviously affected in cKO mice. The data was included in the revised Extended Data Fig. 8j.

We are also puzzled by the “equal numbers of PECAM1⁺ cells but compromised expression of VE-cadherin at E12.5”. A recent study indicated that hypoxia condition reduced VE-cadherin levels⁵. Accordingly, TER119⁺ cells fraction was reduced obviously in cKO mice by FACS (Extended Data Fig. 8p). We additionally performed the double immunofluorescent staining of CD71⁶ and TER119, which validated the compromised erythrocyte pool in cKO mice (Extended Data Fig. 8q). These data implied the possible occurrence of hypoxia in cKO mice, which is owed to the limited oxygen supplement by the compromised erythrocyte production. Therefore, we prefer the possible explanation that the vascular structures were exist, but the VE-cadherin levels decreased significantly at E12.5-14.5 in cKO mice, which is probably due to the hypoxia of compromised erythrocytes. However, it still warrants further investigation. The data and discussions were included accordingly (Extended Data Fig. 8p and q; Line 449-452 and 561-568, marked by light green background).

3. Lower magnification images are required for the AGM and FL in order to allow the reader to orient themselves to these tissues. It is impossible to determine where any of the sections shown in ExFig.8i, j, k are located and their tissue of origin. The AGM and FL are very different in structure and orientation so this can be improved by a lower magnification image.

Answer: The lower magnification images were included and reorganized in the revised manuscript (Extended Data Figs. 8i-k). Thank you very much again for your constructive comments and suggestions.

Reference

- 1 Guizzunti, G. & Seemann, J. Mitotic Golgi disassembly is required for bipolar spindle formation and mitotic progression. *Proc Natl Acad Sci U S A* 113, E6590-E6599, (2016).
- 2 Tavares, L. A. et al. CD4 downregulation by the HIV-1 protein Nef reveals distinct roles for the gamma1 and gamma2 subunits of the AP-1 complex in protein trafficking. *J Cell Sci* 130, 429-443, (2017).
- 3 Fazleabas, A. T., Hild-Petito, S. & Verhage, H. G. Secretory proteins and growth factors of the baboon (*Papio anubis*) uterus: potential roles in pregnancy. *Cell Biol Int* 18, 1145-1153, (1994).

- 4 Maier, J. L. et al. The unfolded protein response mediates fibrogenesis and collagen I secretion through regulating TANGO1 in mice. *Hepatology* 65, 983-998, (2017).
- 5 Zolotoff, C., Voirin, A. C., Puech, C., Roche, F. & Perek, N. Intermittent Hypoxia and Its Impact on Nrf2/HIF-1alpha Expression and ABC Transporters: An in Vitro Human Blood-Brain Barrier Model Study. *Cell Physiol Biochem* 54, 1231-1248, (2020).
- 6 Shim, Y. A., Campbell, T., Welivitigoda, A., Dosanjh, M. & Johnson, P. Regulation of CD71(+)TER119(+) erythroid progenitor cells by CD45. *Exp Hematol* 86, 53-66 e51, (2020).

REVIEWER COMMENTS

Reviewer #3 (Remarks to the Author):

The authors have attempted to respond to the concerns raised. They have successfully done so in one area that critiques raised in the last set of reviews. The imaging of the AGM stained by PECAM1 and VE-cadherin along with Runx1 as shown in Ext. Fig.8i and Ext. Fig.8j is improved and our concerns in that area are addressed.

Outstanding issues remain with respect to the quantification of FL HSCs at E15.5 and with the visualization of the FL vasculature at E12.5 and E14.5.

1). The new data showing the FACS plots of E15.5 FL in the WT and cKO are either incorrectly generated or misinterpreted. The plots shown in Fig.6h clearly show that Q3 has a roughly ~13% CD48-CD150+ HSCs in the cKO embryo which is higher than the ~4% in the WT. So the quantification and conclusions that the authors make in Fig.6i cannot be correct because how is the quantified FL HSC count lower in the cKO when the representative FACS plot shows it to be higher? This is a very concerning issue.

2). The images of the FL shown in Ext. Fig8k for E12.5 and E14.5 are not improved. Once again the staining shown in this figure is of VeCadherin but mostly it is just nuclear staining with DAPI and no orientation of the fact that the FL is shown. Also the cKO sections cannot be of the same FL at the same developmental time because they are not continuous and have massive gaps. This could all have been improved by simply showing a larger section of the FL and a counterstain for albumin to label hepatoblasts or cKit to label the hematopoietic progenitors. The VeCadherin staining is also difficult to make out.

Reviewer #4 (Remarks to the Author):

For most parts, I judged that the authors adequately responded to the reviewer's comments, however, I feel there is not sufficient data presented to claim that TANGO6 in mammals is not involved in secretion with the current investigation, and suggest eliminating this argument. Collagen is not typical cargo and requires specialized machinery with

TANGO1 for secretion. If the authors choose to suggest that TANGO6 is not involved in secretion, 1) KD efficiency is not so high with the current investigation, 2) need to show secretion of other cargoes than EGF and collagen I. I would recommend checking bulk secretion in pulse-chase assay.

REVIEWER COMMENTS

Reviewer #3 (Remarks to the Author):

The authors have attempted to respond to the concerns raised. They have successfully done so in one area that critiques raised in the last set of reviews. The imaging of the AGM stained by PECAM1 and VE-cadherin along with Runx1 as shown in Ext. Fig.8i and Ext. Fig.8j is improved and our concerns in that area are addressed.

Outstanding issues remain with respect to the quantification of FL HSCs at E15.5 and with the visualization of the FL vasculature at E12.5 and E14.5.

Thanks very much for your valuable advices. We carefully addressed the remaining concerns. The information was included in the revised manuscript accordingly.

1). The new data showing the FACS plots of E15.5 FL in the WT and cKO are either incorrectly generated or misinterpreted. The plots shown in Fig.6h clearly show that Q3 has a roughly ~13% CD48-CD150+ HSCs in the cKO embryo which is higher than the ~4% in the WT. So the quantification and conclusions that the authors make in Fig.6i cannot be correct because how is the quantified FL HSC count lower in the cKO when the representative FACS plot shows it to be higher? This is a very concerning issue.

Answer: Thanks very much to your careful review. We apologized to the misinterpretation on previous figures. The FACS plots in Fig. 6h indicated the patterns and ratios of LT/ST-HSC in the LSK population. However, previous Fig. 6i indicated the LT/ST HSC ratio in Lin⁻ cells, but not the HSC counts. These results indicated that the LT/ST-HSC proportion in LSK populations was higher, but in Lin⁻ cells was lower in cKO mice than that in the WT controls. We realized that these data were hard to be followed. We think that the populations of Lin⁻ cells were a mixture of hematopoietic progenitor cells, hepatocytes, endothelial cells and etc. To clarify these data, we counted the LT/ST-HSC cells in WT and cKO mice by the FACS assay. The results indicated that the counts of LT/ST-HSC population decreased significantly in cKO mice (3.44 vs 0.89/13.81 vs 6.05 in WT and cKO), which was included in the revised Fig. 6i. However, in the reduced HSPCs pool, the LT/ST-HSC proportion in LSK populations was higher in cKO mice than that in WT. Concordantly, a significantly higher frequency of the G1 phase cells was detected in the LSK cells in the cKO than WT mice (Figs. 6l, m). These data collectively suggested that compromised TANGO6 blocked the cell cycle of LT/ST-HSCs, which might be quiescently maintained but did not efficiently produce hematopoiesis as those in their WT counterparts. We retrospectively studied previous studies and found that the BRD4 mutant (another hematopoietic defective allele) published in the EMBO J presented a similar phenotype¹ (see below, Fig.1). The ratio of LSK population in Lin⁻ cells was decreased (4.62% vs 0.44%), but the proportion of HSC in LSK cells were increased (29.2% vs 48.8%) in the BRD4 cKO mice than WT. And the counts of HSC were decreased in BRD4 cKO mice at the same time (see red box). We suspected that the HSC ratio phenotypes might be the proportion issues in the mutants with severely compromised hematopoiesis. This phenotype was very interesting and warranted future investigation. The revised data and

discussions were included accordingly (Fig. 6i; Line 442-445 and 577-583, marked by gray background).

Fig1. The phenotype analysis of BRD4 cKO mice¹

2). The images of the FL shown in Ext. Fig8k for E12.5 and E14.5 are not improved. Once again the staining shown in this figure is of VeCadherin but mostly it is just nuclear staining with DAPI and no orientation of the fact that the FL is shown. Also the cKO sections cannot be of the same FL at the same developmental time because they are not continuous and have massive gaps. This could all have been improved by simply showing a larger section of the FL and a counterstain for albumin to label hepatoblasts or cKit to label the hematopoietic progenitors. The VeCadherin staining is also difficult to make out.
 Answer: Thank you for your valuable advices. In our revised manuscript, we imaged a larger section of VE-cadherin and PECAM1 at E12.5-E14.5 as suggested. The immunofluorescent staining images indicated that VE-cadherin protein levels decreased obviously while PECAM1 protein level remained unchanged in the cKO mice when compared with WT mice (Extended Data Figs. 8k and l). Albumin and c-Kit or Runx1 (another classical marker of hematopoietic stem/progenitor cells) were examined at E14.5 accordingly. The immunofluorescent staining images indicated that Albumin⁺, c-Kit⁺ and Runx1⁺ signals reduced significantly in the cKO mice. The data and discussions were included (Extended Data Figs. 8o and p; Line 438-440 and 573-577, marked by gray background). Thank you very much again for your constructive comments and suggestions.

Reviewer #4 (Remarks to the Author):

For most parts, I judged that the authors adequately responded to the reviewer's comments, however, I feel there is not sufficient data presented to claim that TANGO6 in mammals is not involved in secretion with the current investigation, and suggest eliminating this argument. Collagen is not typical cargo and requires specialized machinery with TANGO1 for secretion. If the authors choose to suggest that TANGO6 is not involved in secretion, 1) KD efficiency is not so high with the current investigation, 2) need to show secretion of other cargoes than EGF and collagen I. I would recommend checking bulk secretion in pulse-chase assay.

Answer: Thank you very much for your valuable and constructive suggestions. We

performed the pulse-chase assay to check bulk secretion in *TANGO6*^{WT} and *TANGO6*^{KO} cells as suggested. Briefly, we selected stable isotope [2,3-¹³C₂]alanine to label proteins and followed their secretion. Firstly, we used cy5-alanine to treat *TANGO6*^{WT} and *TANGO6*^{KO} cells. The immunofluorescent staining results indicated that alanine was well contained by both *TANGO6*^{WT} and *TANGO6*^{KO} cells. Then, we performed the pulse chase assay on these cells². The relative abundance of [2,3-¹³C₂]alanine gradually reduced in the cell pellet and increased in the medium supernatant in the WT cell samples. Interestingly, *TANGO6*^{KO} cells presented a much faster decline in cell pellet but quicker ascend in the medium supernatant during the first 2 hours, when compared to WT counterparts. This result indicated that the secretion was not compromised but even enhanced in the *TANGO6*^{KO} samples. The possible enhanced secretary abilities in *TANGO6*^{KO} cells are a very interesting phenotype warranting future investigation. The data and discussions were included accordingly (Extended Data Figs. 3n-p; Line 192-212 and 514-517, marked by light green background). Thank you again for your valuable advice!

Reference

1. Dey A, Yang W, Gegonne A, Nishiyama A, Pan R, Yagi R, Grinberg A, Finkelman FD, Pfeifer K, Zhu J, Singer D, Zhu J, Ozato K. BRD4 directs hematopoietic stem cell development and modulates macrophage inflammatory responses. *EMBO J.* 2019 Apr 1;38(7):e100293.
2. Simon E, Kornitzer D. Pulse-chase analysis to measure protein degradation. *Methods Enzymol.* 2014;536:65-75.

REVIEWER COMMENTS

Reviewer #3 (Remarks to the Author):

The authors have resolved all the concerns with the sections and imaging of the fetal liver. Extended Fig.8o and p clearly show the albumin expressing cells and how the cKit and Runx1 progenitors are reduced in the TANGO6 knockout.

The concerns with the increased proportion of FL HSCs in the TANGO6 knockout still remain. For some reason, the authors continue to insist on a phenotype in the E14.5 fetal liver that is not supported by their data. Rather than reanalyzing their original and poorly assembled flow cytometry dataset, they insist on justifying their conclusions of "decreased numbers but increased proportion of fetal liver HSCs" in the TANGO6 knockout by providing examples of other published work that is not similarly gating for SLAM populations. The reasoning behind these conclusions is flawed because it is based on data that likely contains other contaminating populations in their E15.5 fetal liver HSC plot shown in Fig.6h. This data shows a large number of cells on the x-axis for both WT and cKO fetal livers. As presented by the authors, this is not a correctly gated population. Please see a number of examples of the proper SLAM-HSC gate by several articles, specifically the Kiel MJ et al 2005 Cell (Fig.4-5) and Acar M et al 2015 Nature (Fig.1). The gating in these papers shows how there should be no excess population of cells on the x-axis and only then will the cells in Quadrant 3 be considered to be genuine long-term HSCs. Thus the argument by the authors of why their HSC proportion is high is not valid and the data presented in Fig.6h does not support their conclusion. The absolute counts of HSCs are also based on these flow cytometry gating, so once again, this needs to be reanalyzed and carefully gated for proper SLAM-HSCs in order to make a solid conclusion of what is happening to the frequency and counts of fetal liver HSCs in the knockout mouse.

Reviewer #3 (Remarks to the Author):

The authors have resolved all the concerns with the sections and imaging of the fetal liver. Extended Fig.8o and p clearly show the albumin expressing cells and how the cKit and Runx1 progenitors are reduced in the TANGO6 knockout.

Thanks very much for your great efforts and comments. We addressed the remaining concerns as suggested. The information was included in the revised manuscript accordingly.

The concerns with the increased proportion of FL HSCs in the TANGO6 knockout still remain. For some reason, the authors continue to insist on a phenotype in the E14.5 fetal liver that is not supported by their data. Rather than reanalyzing their original and poorly assembled flow cytometry dataset, they insist on justifying their conclusions of "decreased numbers but increased proportion of fetal liver HSCs" in the TANGO6 knockout by providing examples of other published work that is not similarly gating for SLAM populations. The reasoning behind these conclusions is flawed because it is based on data that likely contains other contaminating populations in their E15.5 fetal liver HSC plot shown in Fig.6h. This data shows a large number of cells on the x-axis for both WT and cKO fetal livers. As presented by the authors, this is not a correctly gated population. Please see a number of examples of the proper SLAM-HSC gate by several articles, specifically the Kiel MJ et al 2005 Cell (Fig.4-5) and Acar M et al 2015 Nature (Fig.1). The gating in these papers shows how there should be no excess population of cells on the x-axis and only then will the cells in Quadrant 3 be considered to be genuine long-term HSCs. Thus the argument by the authors of why their HSC proportion is high is not valid and the data presented in Fig.6h does not support their conclusion. The absolute counts of HSCs are also based on these flow cytometry gating, so once again, this needs to be reanalyzed and carefully gated for proper SLAM-HSCs in order to make a solid conclusion of what is happening to the frequency and counts of fetal liver HSCs in the knockout mouse.

Answer: We deeply appreciate your careful estimation and kind suggestions. We apologize for the mistakes in the previous raw data analysis, which gave contaminating cells on the x-axis and flawed the results. We re-analyzed the previous data as suggested and found that we did not properly set the models of fluorescence compensation and axis display. We tried to improve the setting. However, our old FACS machine (Beckman MoFlo XDP) merely contains three lasers (488/561/647 nm) and is quite difficult to properly set up the compensation. Therefore, we repeated these experiments using new samples in a full spectrum flow cytometry (Cytek® Northern Lights). This machine uses a large number of detectors with narrow band-pass filters to measure a fluorophore's signal, which improved the data quality. We sorted LSK cells and SLAM-HSCs by a proper fluorescence compensation and PMT voltage. We selected logicle (biex) to display the flow cytometry data and counted the ratio of LSK and SLAM-HSCs in the whole sorting cells (total fetal liver cells) (referring to Acar M et al 2015, Nature), which factually indicated the data. The results indicated that both LSK and SLAM-HSCs fractions decreased obviously in E15.5 cKO mice than that in WT controls (LSK: WT, 0.42% ±

0.018%; cKO, 0.016% \pm 0.0018%; SLAM-HSCs: WT, 0.014% \pm 0.00088%; cKO, 0.0027% \pm 0.00023%). Interestingly, the results indicated that cKO mice harbored an additional cluster of c-kit⁺Sca-1⁺ cells than WT controls. We suspected that they are probably the fetal liver cells expressing high levels of Sca-1 under a special condition, like hypoxia, in the cKO mice with limited blood, because a previous study reported that hypoxia led to a significant increment of Sca-1 in some stem cells, like mesenchymal stem cell¹. To this end, we performed the immunofluorescent staining. Indeed, the images indicated that the Sca-1⁺ signals increased obviously in the fetal livers of cKO mice than WT controls at E15.5 (please see below). The signature of the Sca-1⁺ cells in cKO mice is an interesting phenomenon deserving further investigation. The data and discussion were included in the revised Figs.6f-i and 6l-m and manuscript (Line 442-445, 460-462, 577-582 and 883-886, marked by gray background). We deeply appreciated your great efforts and constructive comments again!

Fig. The immunofluorescent staining of Sca-1 in the fetal livers of WT and cKO mice.

Reference:

1. A, Khoo CP, Lin WR, Hawa MI, Tropel P, Patrizi MP, Pozzilli P, Alison MR. Hypoxia increases Sca-1/CD44 co-expression in murine mesenchymal stem cells and enhances their adipogenic differentiation potential. *Cell Tissue Res.* 2010 Jul;341(1):111-20. doi: 10.1007/s00441-010-0982-8. Epub 2010 May 23. PMID: 20496083.

REVIEWERS' COMMENTS

Reviewer #3 (Remarks to the Author):

Following a careful reanalysis and reinterpretation of the FL results at E15.5 in the Tango6-null mice, the results shown in Fig.6 f-i are in line with the data indicating pale fetal livers at this time point in development. The text has also been sufficiently modified to reflect these new findings. Finally, the authors have presented a properly assembled flow cytometry plot for SLAM-LSKs.

A final word of caution, though the increase in Sca1 levels in the FL seems intriguing, it is important to point out that Sca1 on its own is not a hematopoietic marker and labels mouse progenitors in several tissues. Though interesting, it is very likely that this increase of Sca1 levels in the Tango-null FL is non-specific.

Reviewer #3 (Remarks to the Author):

Following a careful reanalysis and reinterpretation of the FL results at E15.5 in the Tango6-null mice, the results shown in Fig.6 f-i are in line with the data indicating pale fetal livers at this time point in development. The text has also been sufficiently modified to reflect these new findings. Finally, the authors have presented a properly assembled flow cytometry plot for SLAM-LSKs.

A final word of caution, though the increase in Sca1 levels in the FL seems intriguing, it is important to point out that Sca1 on its own is not a hematopoietic marker and labels mouse progenitors in several tissues. Though interesting, it is very likely that this increase of Sca1 levels in the Tango-null FL is non-specific.

Thanks very much for your great efforts and constructive suggestion again. We agreed with your opinion and will extend the investigation in the future.